# Hemimetabolous insects elucidate the origin of sexual development via alternative splicing

Judith Wexler[1]*, Emily Kay Delaney[1], Xavier Belles[2], Coby Schal[3],
Ayako Wada-Katsumata[3], Matthew J Amicucci[4], Artyom Kopp[1]

[1]Department of Evolution and Ecology, University of California, Davis, Davis, United States; [2]Institut de Biologia Evolutiva, Consejo Superior de Investigaciones Cientificas, Universitat Pompeu Fabra, Barcelona, Spain; [3]Department of Entomology and Plant Pathology, North Carolina State University, Raleigh, United States; [4]Department of Chemistry, University of California, Davis, Davis, United States

*For correspondence:
jrwexler@ucdavis.edu

Present address: †Department of Entomology, University of Maryland, College Park, United States

Competing interests: The authors declare that no competing interests exist.

**Abstract** Insects are the only known animals in which sexual differentiation is controlled by sex-specific splicing. The *doublesex* transcription factor produces distinct male and female isoforms, which are both essential for sex-specific development. *dsx* splicing depends on *transformer*, which is also alternatively spliced such that functional Tra is only present in females. This pathway has evolved from an ancestral mechanism where *dsx* was independent of *tra* and expressed and required only in males. To reconstruct this transition, we examined three basal, hemimetabolous insect orders: Hemiptera, Phthiraptera, and Blattodea. We show that *tra* and *dsx* have distinct functions in these insects, reflecting different stages in the changeover from a transcription-based to a splicing-based mode of sexual differentiation. We propose that the canonical insect *tra-dsx* pathway evolved via merger between expanding *dsx* function (from males to both sexes) and narrowing *tra* function (from a general splicing factor to dedicated regulator of *dsx*).
DOI: https://doi.org/10.7554/eLife.47490.001

## Introduction

Sex determination and sexual differentiation evolve on drastically different time scales. Sex determination (that is, the primary signal directing the embryo to develop as male or female) can be either environmental or genetic. Genetic sex determination can in turn be either male- or female-heterogametic (with or without heteromorphic sex chromosomes), mono- or polygenic, or haplo-diploid, among other mechanisms (*Charlesworth, 1996*; *Graves, 2006*; *Ellegren, 2000*; *Jablonka and Lamb, 1990*; *Ser et al., 2010*; *Beye et al., 2003*; *Merchant-Larios and Díaz-Hernández, 2013*). Primary sex determination is among the most rapidly evolving developmental processes. In African clawed frogs, medaka, salmon, and other animals, there are many examples of recently evolved sex-determining genes, so that the primary sex determination genes can differ between closely related species (*Bewick et al., 2011*; *Roco et al., 2015*; *Myosho et al., 2012*; *Takehana et al., 2008*; *Li et al., 2011*; *Tree of Sex Consortium et al., 2014*). The signals initiating male or female development can vary even within species, as seen in cichlids, zebrafish, house flies, and ranid frogs (*Ser et al., 2010*; *Anderson et al., 2012*; *Liew et al., 2012*; *Tomita and Wada, 1989*; *Meisel et al., 2016*; *Nakamura, 2009*).

In contrast, sexual differentiation (that is, the set of molecular mechanisms that translates the primary sex-determining signal into sexually dimorphic development of specific organs) tends to remain stable for hundreds of millions of years despite the rapid evolution of both sex determination and

the multitude of anatomical, physiological, and behavioral manifestations of sexual dimorphism. This fits the broader 'hourglass' pattern of developmental evolution, where the most upstream and downstream tiers of developmental hierarchies diverge more rapidly than the middle tiers (*Duboule, 1994*; *Hanken and Carl, 1996*; *Hazkani-Covo et al., 2005*). In the case of vertebrate sexual development, this middle tier is comprised of sex hormones and their receptors, which are largely conserved from mammals to fish, as are the key components of the network of transcription factors and signaling pathways that toggle gonad development between ovary and testis (*Raymond et al., 1999*; *Brennan and Capel, 2004*; *Hau, 2007*; *Maatouk et al., 2008*; *Matson et al., 2011*; *Lindeman et al., 2015*; *Herpin and Schartl, 2015*). In nematodes, key components of the sexual differentiation pathway are conserved between *Caenorhabditis* and *Pristionchus*, which are separated by 200–300 million years (*Pires-daSilva and Sommer, 2004*).

Similarly, the insect sexual differentiation pathway, while unique among metazoans, shows deep conservation across 300 million years of insect evolution (*Ohbayashi et al., 2001*; *Suzuki et al., 2003*; *Cho et al., 2007*; *Hasselmann et al., 2008*; *Oliveira et al., 2009*; *Saccone et al., 2002*; *Shukla and Palli, 2012a*; *Shukla and Palli, 2012b*). This pathway has three distinctive features. First, the *doublesex* (*dsx*) gene is spliced into a male-specific isoform in males and a female-specific isoform in females; the two isoforms encode transcription factors that share a common N-terminal DNA-binding domain but have mutually exclusive C-termini, which lead them to have distinct and often opposite effects on downstream gene expression and morphological development (*Baker and Wolfner, 1988*; *Burtis and Baker, 1989*; *Baker, 1989*; *McKeown, 1992*; *Coschigano and Wensink, 1993*; *Arbeitman et al., 2004*; *Rideout et al., 2010*). Second, alternative splicing of *dsx* is controlled by the RNA splicing factor *transformer* (*tra*); and third, *tra* itself is alternatively spliced so that it produces a functional protein only in females, while in males a premature stop codon results in a truncated, non-functional Tra protein (*Boggs et al., 1987*; *Inoue et al., 1992*; *Sosnowski et al., 1989*; *Hoshijima et al., 1991*; *Verhulst et al., 2010*). Thus, the male-specific *dsx* isoform is produced by default, while the production of the female *dsx* isoform requires active intervention by *tra*. Despite some differences in details, the *transformer-doublesex* splicing cascade is conserved among insect orders separated by >300 million years, including Diptera, Coleoptera, and Hymenoptera, indicating that this pathway was already present in the last common ancestor of holometabolous insects (*Cho et al., 2007*; *Shukla and Palli, 2012a*; *Shukla and Palli, 2012b*; *Ruiz et al., 2005*). The one exception to this rule is found in Lepidoptera, where *tra* has been secondarily lost and *dsx* splicing is regulated instead by a male-specific protein and a female-specific piRNA (*Kiuchi et al., 2014*).

Although conserved within insects, the *transformer-doublesex* splicing pathway appears to be unique among animals, as nothing similar has been observed in any other animal group. The differences in the molecular basis of sexual differentiation between insects, nematodes, and vertebrates are fundamental. In insects, *dsx* and *tra* operate in a mostly cell-autonomous manner, as evidenced for example by dramatic insect gynandromorphs (*Steinmann-Zwicky et al., 1989*; *Janzer and Steinmann-Zwicky, 2001*; *Robinett et al., 2010*; *Pereira et al., 2010*). In vertebrates, sexual differentiation is largely non-cell-autonomous. During embryonic development, the originally bipotential vertebrate gonad is biased to become either ovary or testis under the influence of the primary sex-determining signal, which can be either genetic or environmental (*Raymond et al., 1999*; *Devlin and Nagahama, 2002*; *Kim and Capel, 2006*; *Shoemaker et al., 2007*). Subsequently, male or female hormones produced by the gonad act through nuclear receptor signaling to direct somatic tissues to differentiate in sex-specific ways. In rhabditid nematodes such as *Caenorhabditis elegans*, a multi-step cell signaling cascade involving secreted ligands, transmembrane receptors, and post-translational protein modification results in sex-specific expression of several transcription factors that direct male or female/hermaphrodite development of particular cell lineages (*Barnes and Hodgkin, 1996*; *Pilgrim et al., 1995*; *Pires-daSilva, 2007*; *Berkseth et al., 2013*).

The deep conservation of fundamentally different mechanisms of sexual differentiation in different animal phyla presents an intriguing question: how do such disparate mechanisms evolve in the first place? Comparison of nematode, crustacean, insect, and vertebrate modes of sexual differentiation suggests that the insect-specific mechanism based on alternative splicing of *dsx* evolved from a more ancient mechanism based on male-specific transcription of an ancestral *dsx* homolog. *dsx* is part of the larger Doublesex Mab-3 Related Transcription factor (DMRT) gene family, which is apparently the only shared element of sexual differentiation pathways across metazoans (*Kopp, 2012*; *Matson and Zarkower, 2012*). *Dmrt1* is involved in the specification and maintenance of testis fate

in mammals and other vertebrates, while repressing ovarian differentiation (*Matson et al., 2011*; *Smith et al., 2003*; *Webster et al., 2017*). Due to their function in the androgen-producing Sertoli cells, vertebrate *Dmrt1* genes also play a crucial role in the development of secondary sexual characters. In *C. elegans*, *mab-3* and several other DMRT family genes are responsible for the specification of various male-specific cells, structures, and behaviors (*Shen and Hodgkin, 1988*; *Portman, 2007*; *Mason et al., 2008*; *Siehr et al., 2011*; *Serrano-Saiz et al., 2017*). However, in both vertebrates and nematodes, the DMRT genes involved in sexual differentiation are not spliced sex-specifically, but are transcribed in a predominantly male-specific fashion, promote the development of male-specific traits, and are dispensable for female sexual differentiation. In contrast, the insect *dsx* acts as a bimodal switch that plays active roles in both male- and female-specific differentiation through its distinct male and female splicing isoforms. In the absence of Dsx, both males and females develop as intersexes with phenotypes intermediate between males and females (*Rideout et al., 2010*; *Baker and Ridge, 1980*).

In the closest relative of insects that has been studied to date, the branchiopod crustacean *Daphnia magna*, *dsx* acts in a manner similar to vertebrates and nematodes rather than insects: it is transcribed male-specifically, is not alternatively spliced, controls the development of male-specific structures, and is dispensable in females (*Kato et al., 2011*). The *D. magna tra* gene is not spliced sex-specifically and does not differ in expression between males and females (*Kato et al., 2010*). *dsx* also shows male-biased transcription and no sex-specific splicing in the shrimp *Fenneropenaeus chinensis* (*Li et al., 2018*) and in the mite *Metaseiulus occidentalis* (*Pomerantz and Hoy, 2015*). Thus, the origin of the *transformer-doublesex* splicing cascade was a key event in the evolution of sexual differentiation in insects.

To understand the evolutionary transition from a transcription-based to a splicing-based mode of sexual differentiation, it is necessary to focus on the phylogenetic interval between branchiopod crustaceans and holometabolous insects. This interval spans several crustacean groups, non-insect hexapods, and many hemimetabolous insect orders, and encompasses a broad range of body plans and developmental mechanisms. Unfortunately, the development of these groups remains poorly studied compared to holometabolous insects; in particular, very little is known about sexual differentiation in hemimetabolous insects.

To explore the origin of the insect sexual differentiation pathway based on the *transformer-doublesex* axis, we examined the expression of these genes in three hemimetabolous insects: the kissing bug *Rhodnius prolixus* (Hemiptera), the louse *Pediculus humanus* (Phthiraptera), and the German cockroach *Blattella germanica* (Blattodea) (*Figure 1—figure supplement 1*). We find that distinct male and female *dsx* isoforms are conserved through Blattodea, the most basal insect order studied to date. However, only *R. prolixus* shows the canonical sex-specific pattern of *tra* splicing. In *B. germanica*, we show that *tra* nevertheless controls female *dsx* splicing and is necessary for female sexual differentiation, as in holometabolous insects. Surprisingly, the *B. germanica dsx* gene is required for male sexual differentiation, but appears to be dispensable in females; in this respect, the cockroach is more similar to crustaceans and non-arthropod animals than to holometabolous insects. Together, our results suggest that the splicing-based mode of sexual differentiation based on the *transformer-doublesex* regulatory pathway has evolved in a gradual fashion in hemimetabolous insects.

## Results

### *doublesex* and *transformer* orthologs are present in hemimetabolous insects

Although *dsx*, *Dmrt1*, and other DMRT genes have prominent roles in sexual differentiation (*Raymond et al., 1999*; *Burtis and Baker, 1989*; *Kopp, 2012*; *Matson and Zarkower, 2012*; *Shen and Hodgkin, 1988*) many DMRT paralogs are involved in other developmental processes, and most have phylogenetically restricted distributions (*Wexler et al., 2014*). The number of DMRT paralogs varies among animal taxa; for example, *Drosophila melanogaster* has four, while *C. elegans* has 11 (*Wexler et al., 2014*). To identify *dsx* orthologs in hemimetabolous insects for functional study, we performed a phylogenetic analysis of arthropod DMRT genes (*Supplementary file 1*). By mining arthropod gene models, we recovered the deeply conserved Dmrt11E, Dmrt93B, and Dmrt99B subfamilies in addition to a large clade containing *dsx* orthologs. The *dsx* clade contained

sequences from several basal insect orders, crustaceans, and chelicerates (*Figure 1—figure supplement 1*), and included experimentally characterized *dsx* genes from the branchiopod crustacean *D. magna* (NCBI accession number BAJ78307.1), the decapod crustacean *F. chinensis* (AUT13216.1), and the mite *M. occidentalis* (XP003740429.2 and XP003740430.1). In these species, the *dsx* genes are not spliced sex-specifically, but are transcribed in a strongly male-biased fashion (*Kato et al., 2011*; *Li et al., 2018*; *Pomerantz and Hoy, 2015*). *dsx* genes from holometabolous insects, which direct male and female differentiation via alternative splice forms, are also present in this clade. There are marked departures in this gene tree from the arthropod species tree. Notably, many hemipteran *dsx* sequences group with chelicerates and crustaceans instead of insects, although with low support. The clustering of crustacean and chelicerate *dsx* genes, which control male sexual development via male-specific upregulation, with holometabolous *dsx* genes that control male and female development via alternative spliceforms indicates that sex-specific *dsx* isoforms evolved after the origin of the *dsx* clade.

Puzzled by the clustering of hemipteran *dsx* genes with those from chelicerates and crustaceans, we conducted a synteny analysis. While the robustness of this analysis is dependent on the quality of genome assembly, the transcription factor *prospero (pros)* is on the same scaffold as *dsx,* with an intervening distance between 17 and 245 kb (*Supplementary file 2*), in seven different holometabolous and hemimetabolous insect orders (Ephemeroptera, Blattodea, Phthiraptera, Thysanoptera, Hymenoptera, Coleoptera and Lepidoptera). However, in Hemiptera, tBLASTn searches of the *prospero*-containing scaffolds, ranging in size from ~120 kb to 17 Mb (mean 2.75 Mb), did not identify neighboring DMRT genes (*Supplementary file 3*). Work in the planthopper *Nilaparvata lugens* suggests that at least some of the hemipteran *dsx* genes are alternatively spliced and are necessary for male development (*Zhuo et al., 2018*). We conclude that hemipterans have *dsx* orthologs, but the synteny between *pros* and *dsx* was lost in the common ancestor to true bugs, and Dsx protein sequences evolved rapidly in this clade.

Transformer proteins have repetitive sequences that evolve rapidly, posing greater difficulties for phylogenetic analysis. Previous studies of insect *tra* genes have identified these genes via the characteristic arginine/serine rich (RS) domain (*Hasselmann et al., 2008*), used BLAST to detect a *tra* candidate and demonstrated congruence between a species tree and a gene tree (*Kato et al., 2010*), or simply concluded that a candidate *tra* gene was indeed *transformer* after knocking it down and obtaining a sex-specific phenotype (*Shukla and Palli, 2012a*). *tra* is related to, but is not part of, a large family of genes structurally defined by the presence of an RNA binding domain and at least one RS domain (*Long and Caceres, 2009*). These proteins, called SR family genes for their amino acid composition, commonly function in pre-mRNA splicing (*Long and Caceres, 2009*). Because most insect *tra* genes lack an RNA binding domain (see below), they are classified as RS-like proteins. Our maximum likelihood phylogenetic analysis of Tra proteins, along with three previously studied SR family genes (Transformer-2, which is not an ortholog of Transformer; SFRS; and Pinin) shows that putative *tra* genes from some hemimetabolous insects including *R. prolixus, P. humanus*, and *B. germanica* cluster with experimentally characterized holometabolous *tra* genes and with the *D. magna tra*, although with low support (*Figure 1—figure supplement 1*). Despite poor phylogenetic resolution, the putative *tra* genes identified in *R. prolixus*, *P. humanus,* and *B. germanica* provided an avenue for experimental analysis of sexual differentiation in hemimetabolous insects.

Prior work has shown that *dsx* is arthropod-specific (*Wexler et al., 2014*); the present analyses confirm that *dsx* was probably present in the last common ancestor of arthropods. More research is needed to pinpoint the origin of *tra*, but the poorly conserved and highly repetitive nature of Tra proteins suggests that this problem might be intractable.

## *tra* splicing follows the holometabolous sex-specific pattern in *R. prolixus* but not in *P. humanus* or *B. germanica*

Insect Tra proteins are defined by three key features: (1) an arginine-serine rich (RS) domain, (2) a proline rich domain, and, (3) for all non-*Drosophila tra* sequences, an auto-regulatory CAM domain named for the three genera in which it was discovered, *Ceratitis-Apis-Musca* (*Verhulst et al., 2010*; *Hediger et al., 2010*; *Tanaka et al., 2018*). In the holometabolous insect orders Diptera, Coleoptera, and Hymenoptera, a premature male-specific stop codon truncates the *tra* coding sequence, making the male Tra protein unable to regulate *dsx* splicing (*Hasselmann et al., 2008*; *Boggs et al., 1987*; *Inoue et al., 1992*; *Sosnowski et al., 1989*). These insects produce a male-specific *dsx*

isoform by default, while females produce a female-specific *dsx* isoform as a result of having functional Tra. To understand when *tra*-dependent alternative splicing of *dsx* evolved, we identified *tra* isoforms expressed in sexually dimorphic tissues of males and females from three hemimetabolous insect orders (Hemiptera: *R. prolixus*; Phthiraptera: *P. humanus*; and Blattodea: *B. germanica*). We find that the characteristic holometabolous-like pattern of sex-specific splicing of *tra*, with a male-specific premature stop codon, is present in *R. prolixus* but not in *P. humanus* or *B. germanica*. Instead, *P. humanus* and *B. germanica* display novel patterns of alternative splicing, which include the presence of female-biased truncated *tra* isoforms and an interrupted CAM domain.

## *Rhodnius prolixus*

In R. prolixus, we identified a tandem duplication of *tra* on scaffold KQ034193 (Rhodnius-prolixus-CDC_SCAFFOLDS_RproC3, v.1) via tBLASTn searches of the R. prolixus genome (*Mesquita et al., 2015*). In the phylogenetic tree of arthropod SR proteins (*Figure 1—figure supplement 1*), the two R. prolixus paralogs (*RpTraA* and *RpTraB*) cluster within a clade that also contains dipteran, hymenopteran, and coleopteran Tra proteins. The downstream *R. prolixus tra* paralog, which we call *RpTraB*, encodes conserved Tra amino acid residues at its N-terminus, including the CAM domain and an RS domain, but is truncated at the C-terminus and lacks a proline-rich domain (*Figure 1—figure supplement 2*). In contrast, the upstream *RpTraA* paralog encodes a full-length Tra protein with a C-terminal proline-rich domain. Interestingly, *RpTraA* is also predicted to contain a partial RNA Recognition Motif (RRM) (*Figure 1—figure supplement 3*), a feature which has not been described in previous studies of Tra proteins. However, when we inspected predicted protein domains in Tra sequences from Branchiopoda, Copepoda, Blattodea, Hemiptera, Hymenoptera, Coleoptera, and Diptera with CCD/SPARCLE software via NCBI (*Marchler-Bauer et al., 2017*), we found putative RRM domains in *Apis mellifera* CSD, *R. prolixus* TraA, a predicted Tra protein from the copepod crustacean *Tigriopus californicus*, and the cockroach *B. germanica* Tra ortholog (*Figure 1—figure supplement 3*). The absence of the RRM domain from most arthropod Tra sequences suggests that insect Tra proteins once had a functional RRM domain but lost RNA-binding activity over time, perhaps due to the association between Tra and the RNA-binding protein Tra-2 (48). RpTraA and RpTraB are the first reported tra duplicates outside of Hymenoptera, where tra duplications sometimes result in functional innovation (*Hasselmann et al., 2008*; *Geuverink and Beukeboom, 2014*).

To investigate whether *RpTraA* is expressed similarly to holometabolous *tra* orthologs, we conducted 5′ and 3′ Rapid Amplification of cDNA Ends (RACE) to obtain the sets of isoforms expressed in male and female R. prolixus. Using separate male and female RNA samples, we also generated de novo transcriptome assemblies using Illumina sequencing and Trinity software (*Grabherr et al., 2011*). We were unable to recover *RpTraB* from adult male or female gonad transcriptomes, nor could we recover any *RpTraB* product by RT-PCR. We found two female-specific *RpTraA* transcripts (*RpTraA_3* and *RpTraA_4*) and two male-specific transcripts (*RpTraA_1* and *RpTraA_2*) (*Figure 1A*). All transcripts were identified as sex-specific based on their presence/absence in male and female RACE and RNA-seq libraries, and their sex-specificity was further confirmed by RT-PCR (*Figure 1B*). As in the holometabolous *tra* splicing, both male-specific transcripts (*RpTraA_1* and *RpTraA_2*) contain stop codons near the N-terminus and thus encode truncated proteins that are almost certainly incapable of regulating RNA splicing (*Figure 1A*). Only the *RpTraA_4* transcript encodes a complete Tra protein with the CAM domain followed by an RS domain, RRM domain, and a proline-rich domain (*Figure 1A*). The female-specific *RpTraA_3* lacks the stop codon found in male *RpTraA* transcripts, but has truncated RS and CAM domains and lacks the proline-rich domain. The mapping of male and female RNA-seq reads to the four *RpTraA* isoforms identified by RACE further confirmed the sex-specificity of these isoforms: no female reads mapped to the male-specific stop codons in *RpTraA_1* and *RpTraA_2*, and no male reads mapped to female-specific exon junctions of *RpTraA_3* and *RpTraA_4* (*Figure 1—figure supplement 4*). This pattern – a male-specific premature stop codon that is located in an alternatively spliced exon and results in the production of full-length Tra protein in females but not in males – follows the same pattern as in holometabolous insects including *D. melanogaster*, *A. mellifera*, and *Tribolium castaneum* (*Hasselmann et al., 2008*; *Shukla and Palli, 2012a*; *Sosnowski et al., 1989*). Our detection of truncated Tra isoforms in males but not females in a hemipteran suggests that at least some elements of the sexual differentiation mechanism based on the alternative splicing of *tra* may predate holometabolous insects.

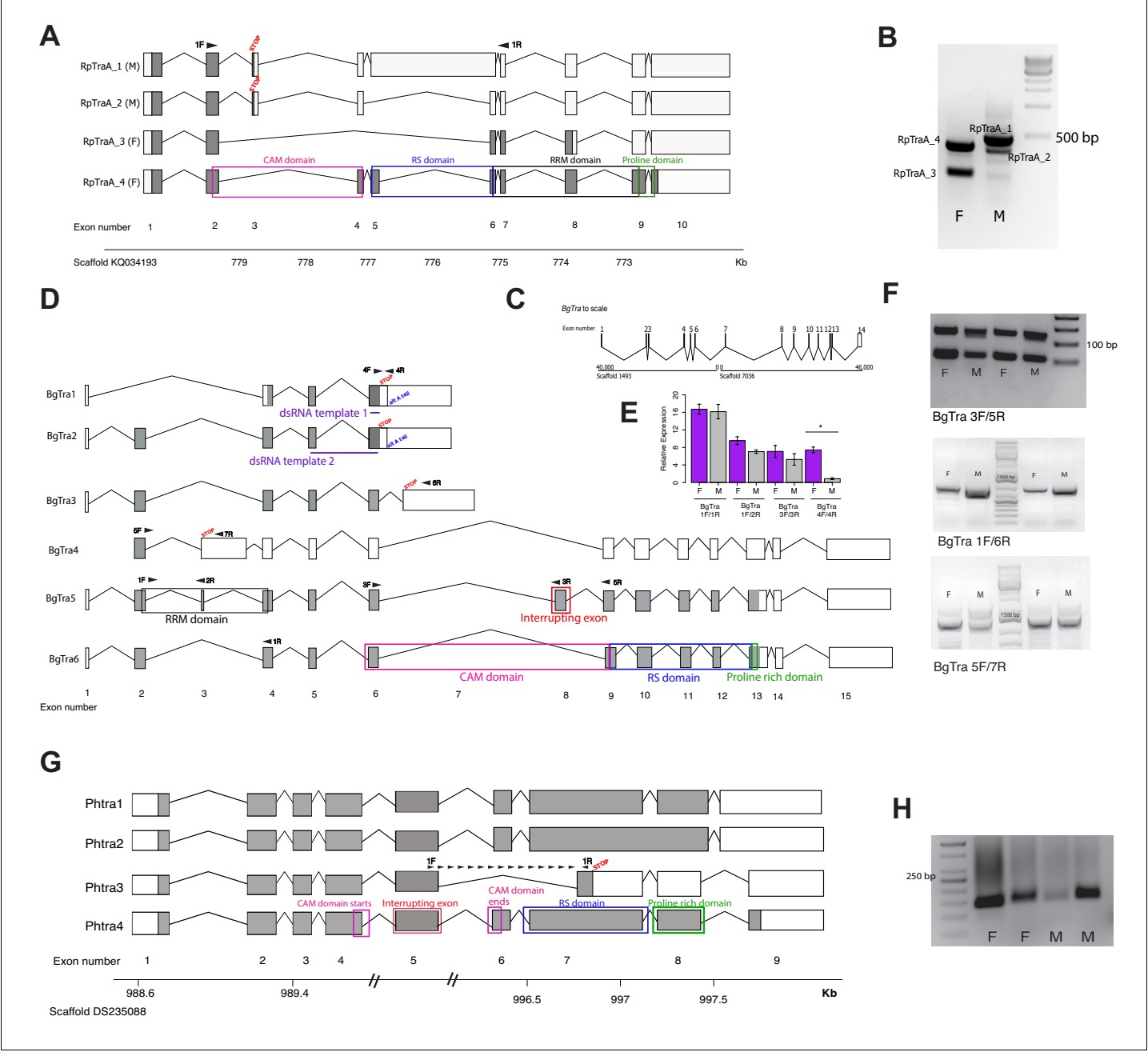

**Figure 1.** Transformer splicing follows the holometabolous pattern in the kissing bug *Rhodnius prolixus*, but not in the cockroach *Blattella germanica* or louse *Pediculus humanus*. Schematics showing *transformer* transcripts isolated from *R. prolixus*, *B. germanica*, and *P. humanus*. Coding sequences are in gray, UTRs are in white. All genes are shown as transcribed from left to right. (**A**) Four transcripts isolated from *RpTraA* with 5' and 3' RACE and confirmed by RNA-seq. Sex-specificity of each transcript is noted in parentheses at the end of the transcript name. Stop codon in the male-specific 3rd exon is indicated with 'STOP.' Predicted Transformer protein domains are shown with colored boxes and labels, using *RpTraA_4* as example. (**B**) PCR on male and female cDNA with *RpTraA* primers (RpTra_checkF + RpTra_checkR in *Supplementary file 7*) indicated by black arrows in (**A**) confirms the sex-specificity of each transcript; all amplicons were verified by Sanger sequencing. (**C**) *BgTra* spans two scaffolds in the *B. germanica* genome. Genome coordinates are shown at the beginning and end of each scaffold. Both scaffolds continue beyond *BgTra*; scaffold 1493 is 552 kb and scaffold 7036 is 55 kb. (**D**) Six *BgTra* transcripts were recovered in PacBio Isoseq data and verified by exon-specific RT-PCR. *BgTra1*, *BgTra2* and *BgTra4* were recovered from the female Isoseq dataset; *BgTra3*, *BgTra5*, and *BgTra6* were recovered from the male Isoseq dataset; however, PCR and qPCR on male and female cDNA showed the presence of *BgTra3-6* in both sexes (**E, F**). Nucleotides that code for important protein domains are color-coded as in panel A, using *BgTra5* and *BgTra6* as examples. *BgTra1* and *BgTra2* have alternative polyadenylation sites, marked with blue text. In *BgTra1* and *BgTra2*, exon six extends ~500 bp further 3' than in all other transcripts. The 3' end of exon six in *BgTra3-6* falls six bp upstream of the stop codon that ends the conceptual translation of *BgTra1* and *BgTra2*. Exon 8 codes for 47 amino acids that interrupt the CAM domain, which spans the end of exon

*Figure 1 continued on next page*

*Figure 1 continued*

six and the start of exon 9. Purple lines show the locations of dsRNA used for RNAi knock-down. Black arrows indicate locations of primers used for RT-PCR and qPCR. (E) qPCR expression of different *BgTra* isoforms. The truncated transcripts *BgTra1* and *BgTra2* are strongly female-biased, while the remaining transcripts are sexually monomorphic. (F) RT-PCR expression of different *BgTra* isoforms. Top panel: both males and females express transcripts with and without the exon interrupting the CAM domain. The shorter band contains *BgTra* isoforms that skip exon eight and contain an intact CAM domain; the larger product contains exons 6, 8, and 9. Middle panel: both males and females express exon 7, found only in *BgTra3* and producing a truncated transcript. Bottom panel: both males and females express the extended exon three with a premature stop codon found in *BgTra4*. (G) Four *PhTra* transcripts isolated from *P. humanus* by 5' and 3' RACE. *PhTra3* has a premature stop codon and is found in both sexes, as shown by RT-PCR on male and female cDNA in panel (H). Forward primer for PCR shown in (H) spanned the exon 5/7 junction (marked with black arrows in panel (G)); reverse primer for this PCR is in exon 7 (also noted with a black arrow). All *PhTra* isoforms contain the 'interrupting exon' inside of the CAM domain. Double bars on scaffold in panel (G) represent breaks in scaffold scale.

DOI: https://doi.org/10.7554/eLife.47490.002

The following figure supplements are available for figure 1:

**Figure supplement 1.** Three trees: a simplified arthropod phylogeny showing species studied, and gene trees for dsx and tra.

DOI: https://doi.org/10.7554/eLife.47490.003

**Figure supplement 2.** MAFFT protein alignment of RpTraB and the longer female-specific isoform of RpTraA from *Rhodnius prolixus*.

DOI: https://doi.org/10.7554/eLife.47490.004

**Figure supplement 3.** RRM domains found in select Transformer proteins.

DOI: https://doi.org/10.7554/eLife.47490.005

**Figure supplement 4.** RNA-seq supports the sex-specificity of *RpTra* isoforms.

DOI: https://doi.org/10.7554/eLife.47490.006

**Figure supplement 5.** MAFFT protein alignment of the complete CAM domain in nine different insect species and one crustacean.

DOI: https://doi.org/10.7554/eLife.47490.007

## *Blattella germanica*

To characterize *tra* isoforms in male and female German cockroaches, we sequenced full-length transcripts using a combination of PacBio RNA Isosequencing and 5' and 3' RACE on male and female samples. Our tBLASTn search of the *B. germanica* genome revealed a single putative *tra* ortholog (*BgTra*) based on its clustering with *A. mellifera* and *R. prolixus tra* genes (*Figure 1—figure supplement 1C*). Both PacBio RNA Isosequencing and 5'/3' RACE revealed that *BgTra* spans over 86 kb of genomic sequence on two separate scaffolds (genome assembly: i5K Bger Scaffolds v.1) (*Figure 1C*). The gene model of *BgTra* contains all previously described functional domains of insect Tra proteins – a CAM domain, an RS domain, and a proline rich domain (*Figure 1D*). Six transcripts, *BgTra1-6*, were supported by Isoseq and confirmed by Sanger sequencing of RT-PCR products; *BgTra1-2* were further confirmed by 3' RACE, and all but *BgTra4* were confirmed by 5' RACE. Three transcripts (*BgTra1-3*) are truncated and do not extend past the CAM domain, while *BgTra4* contains a premature stop codon (*Figure 1D*). The remaining two transcripts – *BgTra5* and *BgTra6* – encode predicted full-length Tra proteins. *BgTra6* contains an intact CAM domain, while in *BgTra5* this domain is interrupted by an in-frame exon. In addition, *BgTra5* contains a prediced RRM domain (*Figure 1D*).

Unlike holometabolous insects and *R. prolixus*, the production of full-length Tra protein is clearly not limited to females in *B. germanica*. Although *BgTra1*, *BgTra2*, and *BgTra4* were originally isolated by Isoseq from female samples, and *BgTra3*, *BgTra5,* and *BgTra6* from male samples, RT-PCR and qPCR revealed the presence of all isoforms in both sexes (*Figure 1E,F*). Two of the short isoforms with a truncated CAM domain and lacking the RS and proline-rich domains (*BgTra1-2*) have strongly female-biased expression (*Figure 1E*), while another truncated isoform, *BgTra3*, has a slight male bias (*Figure 1F*). Importantly, the two full-length isoforms containing all of the Tra functional domains, *BgTra5-6*, are expressed at sexually monomorphic levels (*Figure 1E,F*).

## *Pediculus humanus*

The louse *P. humanus* shows a pattern of *tra* splicing similar to that in the cockroach. Using a tBLASTn search followed by 5' RACE, we identified a single gene, *PhTra*, in the *P. humanus* genome, which clustered with other *tra* orthologs in our phylogenetic analyses (*Figure 1—figure supplement 1C*). Using 5' and 3' RACE and Illumina RNA-seq followed by de novo transcriptome assembly, we identified four *PhTra* isoforms (*Figure 1G*). *PhTra1* and *PhTra4* were found in both male and female

RACE libraries, as well as in the female RNA-seq assembly. *PhTra2* and *PhTra3* were amplified in only the male RACE library, and were not found in either male or female transcriptome assemblies. *PhTra3* differs from the other isoforms in containing a premature stop codon in the middle of the RS domain (*Figure 1G*). RT-PCR shows that this isoform is present in both males and females (*Figure 1H*). The remaining three *PhTra* isoforms encode full-length proteins including an RS-domain and a proline-rich domain, but contain an exon that interrupts the CAM domain, as in some *BgTra* isoforms (*Figure 1D*, *Figure 1—figure supplement 5*). The location of this exon is conserved between *PhTra* and *BgTra*. In both *tra* orthologs, the CAM domain is interrupted immediately before two highly conserved resides – a glutamic acid followed by a glycine. Remnants of this interrupting exon, which is only found in hemimetabolous *tra* orthologs, are detected in a conserved exon junction right before these two amino acids in the *tra* orthologs of holometabolous insects (*Hediger et al., 2010*).

Overall, our comparison of *R. prolixus*, *B. germanica*, and *P. humanus* shows that the sex-specific splicing of *tra*, which is typical of holometabolous insects and limits Tra protein function to females, is found in some but not all hemimetabolous insects.

## *doublesex* splicing follows the holometabolous, sex-specific pattern in some but not all hemimetabolous insects

In holometabolous insects, the Doublesex transcription factor has male- and female-specific isoforms. The male and female proteins share the DNA-binding DM domain and an oligomerization domain that promotes homodimerization of the transcription factor. This domain is essential to Dsx function, as the protein can only bind DNA efficiently as a dimer (*Erdman et al., 1996*; *Cho and Wensink, 1998*). Male- and female-specific exons at the 3′ end of *dsx* transcripts are responsible for sex-specific expression of Dsx target genes. We recovered one *dsx* ortholog each from *R. prolixus* (*RpDsx*), *P. humanus* (*PhDsx*), and *B. germanica* (*BgDsx*) (*Figure 1—figure supplement 1*). To test for the presence of sex-specific *dsx* isoforms, we isolated *dsx* transcripts using PacBio Isosequencing, Illumina RNA-sequencing, and 5′ and 3′ RACE on sexually dimorphic tissues. In all three species, we identified alternatively spliced *dsx* transcripts that follow the canonical holometabolous pattern characterized by a common N-terminus and mutually exclusive C-termini. Sex-specific *dsx* splicing is conserved through *B. germanica*, although in *P. humanus* both alternatively spliced isoforms are sexually monomorphic.

### Rhodnius prolixus

5′ and 3′ RACE revealed three *RpDsx* isoforms, one of which was isolated from female RNA samples (*RpDsx1*) and two from male samples (*RpDsx2* and R*pDsx3*) (*Figure 2A*). We also retrieved *RpDsx1* from our Trinity de novo transcriptome of R. prolixus female gonads; we were unable to retrieve any *dsx* isoforms from our male R. prolixus transcriptome. RT-PCR confirmed that *RpDsx1* was indeed female-specific, and qPCR showed female-specific expression of *RpDsx1* and male-specific expression of *RpDsx2* and *RpDsx3* (*Figure 2B*). *RpDsx1* and *RpDsx2* had low expression in *R. prolixus* gonads, whereas *RpDsx3* was expressed at a 6- to 8-fold higher level than either *RpDsx1* or *RpDsx2* (*Figure 2B*).

Predicted proteins encoded by all three transcripts share the DM domain at their N-termini. However, we failed to detect an oligomerization domain in any of the *RpDsx* transcripts using either NCBI's CDD/SPARCLE domain predictor software (*Marchler-Bauer et al., 2017*) or EMBL's InterPro (*Mitchell et al., 2019*). We were similarly unable to detect an oligomerization domain in the Dsx protein sequences of another hemipteran – the planthopper *N. lugens* (*Zhuo et al., 2018*) – raising the possibility that hemipteran Dsx has lost this domain and thus has significant functional differences from the Dsx transcription factors of holometabolous insects (*Zhuo et al., 2018*).

### Blattella germanica

With a combination of PacBio isosequencing and RACE, we recovered three *dsx* isoforms from *B. germanica*, each with a different terminal 3′ exon and 5′ and 3′ UTRs (*Figure 2C*). Like *dsx* genes described from the holometabolous orders Hymenoptera, Coleoptera, Lepidoptera, and Diptera (*Suzuki et al., 2003*; *Cho et al., 2007*; *Burtis and Baker, 1989*; *Kijimoto et al., 2012*), these isoforms are identical at their N-termini but differ by alternative splicing at the C-termini. The shared

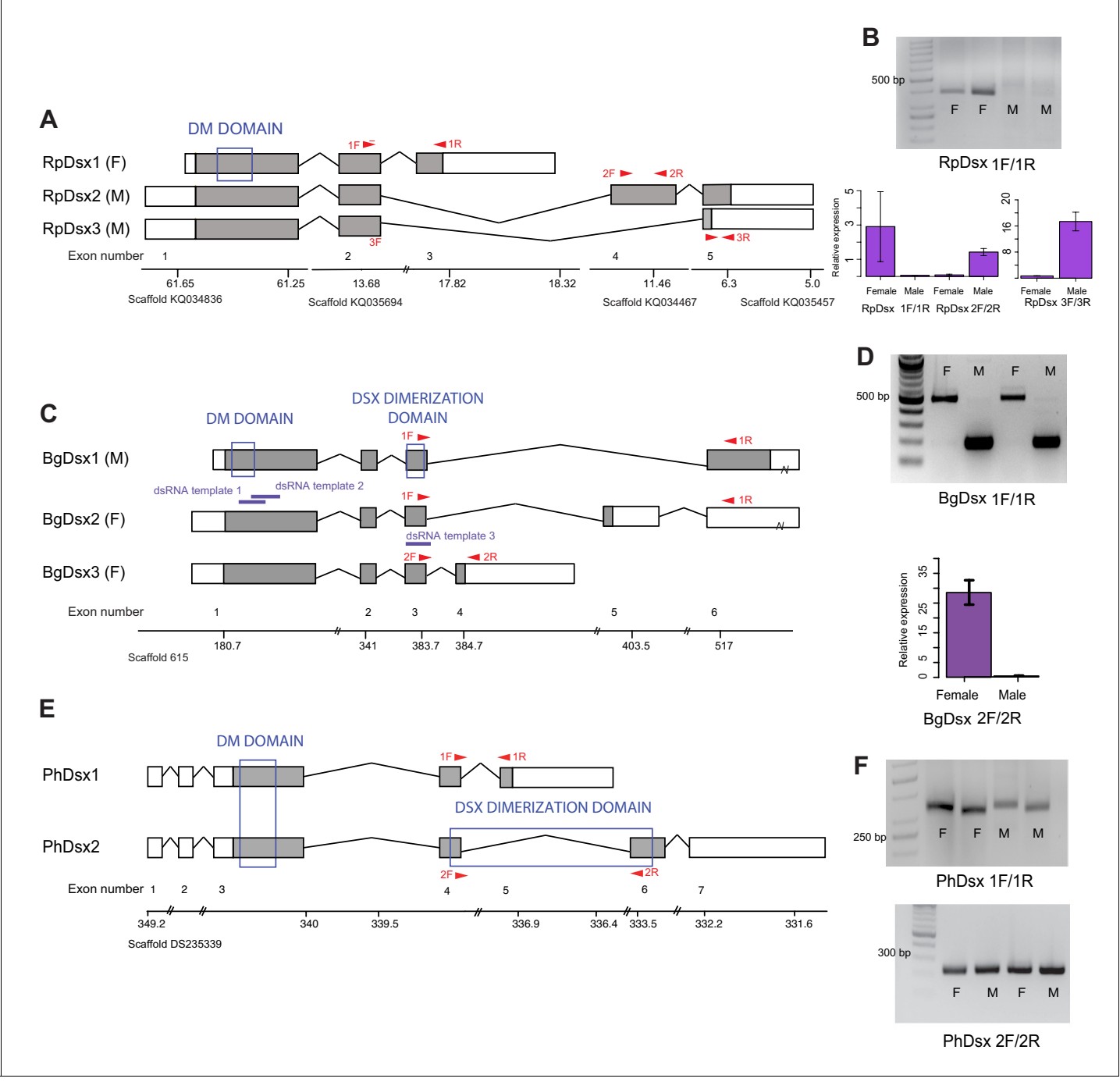

**Figure 2.** *Doublesex* shows sex-specific splicing in some but not all hemimetabolous insects. Schematics showing *doublesex* transcripts isolated from *Rhodnius prolixus*, *Blattella germanica*, and *Pediculus humanus*. Coding sequences are in gray and UTRs in white. Sex-specificity of each transcript is noted in parentheses at the end of the transcript name. Transcripts are shown mapped to genomic scaffolds; double bars show breaks in scale. Functional domains are labeled and indicated by blue boxes. (**A**) Three *dsx* transcripts isolated from *R. prolixus* span multiple genomic scaffolds. *RpDsx1* was isolated from 3' and 5' RACE libraries made from female RNA; *RpDsx2* and *RpDsx3* were cloned from RACE libraries synthesized from male RNA. (**B**) Top panel: RT-PCR showing female-specificity of *RpDsx1*, using primers in exons 2 and 3, as indicated with red arrows in (**A**). Bottom panel: qPCR showing female-specific expression of *RpDsx1*, and male-specific expression of *RpDsx2* and *RpDsx3*, using primers indicated with red arrows in (**A**). Forward primer 3F spanned the exon 2/5 junction. (**C**) Three *dsx* transcripts isolated by 3' and 5' RACE and PacBio Isosequencing from *B. germanica*. The 3' UTR of *BgDsx1* and *BgDsx2* contains a retrotransposon (marked with a break) and cannot be mapped to a single scaffold of the roach genome. Purple lines show the locations of dsRNA used for RNAi. Red arrows show primer locations for RT-PCR and qPCR. (**D**) Sex-specific expression of *BgDsx* isoforms. Top panel, PCR on cDNA of adult male and female fat body and reproductive tract with primers in exons 3 and 6. The larger band found exclusively in females contains the alternatively included exon 5, as confirmed by Sanger sequencing. Bottom panel, qPCR for exon

*Figure 2 continued on next page*

*Figure 2 continued*

3/4 junction shows amplification only in females. (E) Two *dsx* transcripts isolated by 3' and 5' RACE from *P. humanus* male and female samples. (F) RT-PCR shows the presence of both *PhDsx* transcripts in both sexes.

DOI: https://doi.org/10.7554/eLife.47490.008

N-terminal sequence contains both the DM domain and the oligomerization domain (*Figure 2C*). RT-PCR and qPCR revealed that *BgDsx1* is found only in males, while *BgDsx2* and *BgDsx3* are female-specific (*Figure 2D*). This pattern of sexually dimorphic splicing, in which male and female transcripts share the DM domain but have different C-terminal domains, is typical of *dsx* in holometabolous insects (*Cho et al., 2007*; *Shukla and Palli, 2012b*; *Ruiz et al., 2005*).

### *Pediculus humanus*

5' and 3' RACE identified two *PhDsx* isoforms, *PhDsx1* and *PhDsx2*, both of which were found in both male and female RACE libraries (*Figure 2E*). While both *PhDsx1* and *PhDsx2* have DM domains at their N-termini, alternative exon inclusion at the C-terminus yields a predicted Dsx dimerization domain in *PhDsx2* but not *PhDsx1*. RT-PCR on a different set of male and female louse samples confirmed that both *PhDsx1* and *PhDsx2* were expressed in both sexes (*Figure 2F*).

The order Blattodea, which includes cockroaches, is the most basal insect group in which *dsx* splicing has been examined to date. Thus, our results suggest that sex-specific *dsx* splicing appeared early in insect evolution, and the sharing of isoforms between sexes in *P. humanus* may represent a secondary loss of sex-specificity.

### *tra* is necessary for female, but not male development in *B. germanica*

*B. germanica* has a number of overt sexually dimorphic traits. Adult females are darker and have a rounded abdomen, while male abdomens are more slender (*Figure 3A*). Males also have a tergal gland that produces a mixture of oligosaccharides and lipids upon which females feed prior to copulation (*Nojima et al., 1999*). This gland, located on the dorsal abdomen under the wings, appears externally as invagination in the 7th and 8th tergites, resulting in a complex structure with depressions and holes that lead to the internal glands (*Figure 3B*) (*Ylla and Belles, 2015*). Finally, the ovaries, testes, and accessory glands that compose the male and female reproductive systems can be easily distinguished upon dissection (*Figure 3C,D*). To test whether *BgTra*, with its novel splicing pattern, functions to produce male and female traits similarly to its holometabolous orthologs, we used RNAi to knock it down at different stages of development.

In separate experiments, we injected female and male 4th, 5th and 6th instars with two overlapping double-stranded RNA sequences common to all *BgTra* isoforms (*Figure 1B*). *BgTra* expression levels were reduced to similar, very low levels in both males and females despite starting from a higher baseline level in females (*Figure 3—figure supplement 1*). Females treated with *dsBgTra* developed normally until the 6th instar, which took 16–21 days instead of the normal 8 days. Sixth instar female nymphs had an abnormal extrusion of tissue at the level of the last abdominal sternite (*Figure 3—figure supplement 2*). Like wild-type females, *dsBgTra* females had five visible abdominal sternites, but the last sternite had an abnormal morphology with a 'cut away' shape through which soft tissue was visible (*Figure 3—figure supplement 2*). *dsBgTra* female nymphs molted into masculinized adults (*Figure 3*). All *dsBgTra*-treated female nymphs that molted to adults (n = 25) showed the slender abdomen tapered at the posterior end, which is typical of males (*Figure 3A*), and all had well developed tergal glands (*Figure 3B*). To test whether the tergal glands of masculinized females were functional, we extracted the oligosaccharides from the tergal glands of 11 *dsBgTra* females and 10 wild-type adult males and analyzed them by mass spectrometry. All of the 26 oligosaccharides present in the wild-type male tergal glands were also observed in *dsBgTra* female adults; these *BgTra*-depleted adults did not have any additional oligosaccharides not found in wild-type adult males (*Figure 4—figure supplement 1*).

Internally, most adults that molted from the *dsBgTra*-treated female nymphs had underdeveloped ovaries containing large areas of undifferentiated tissue (*Figure 3D*). Some *dsBgTra* females had intersexual gonads ('ovotestes') containing both ovarian and testicular tissue (*Figure 3D*). The collaterial gland, which is located at the base of the abdomen and is responsible for egg case protein

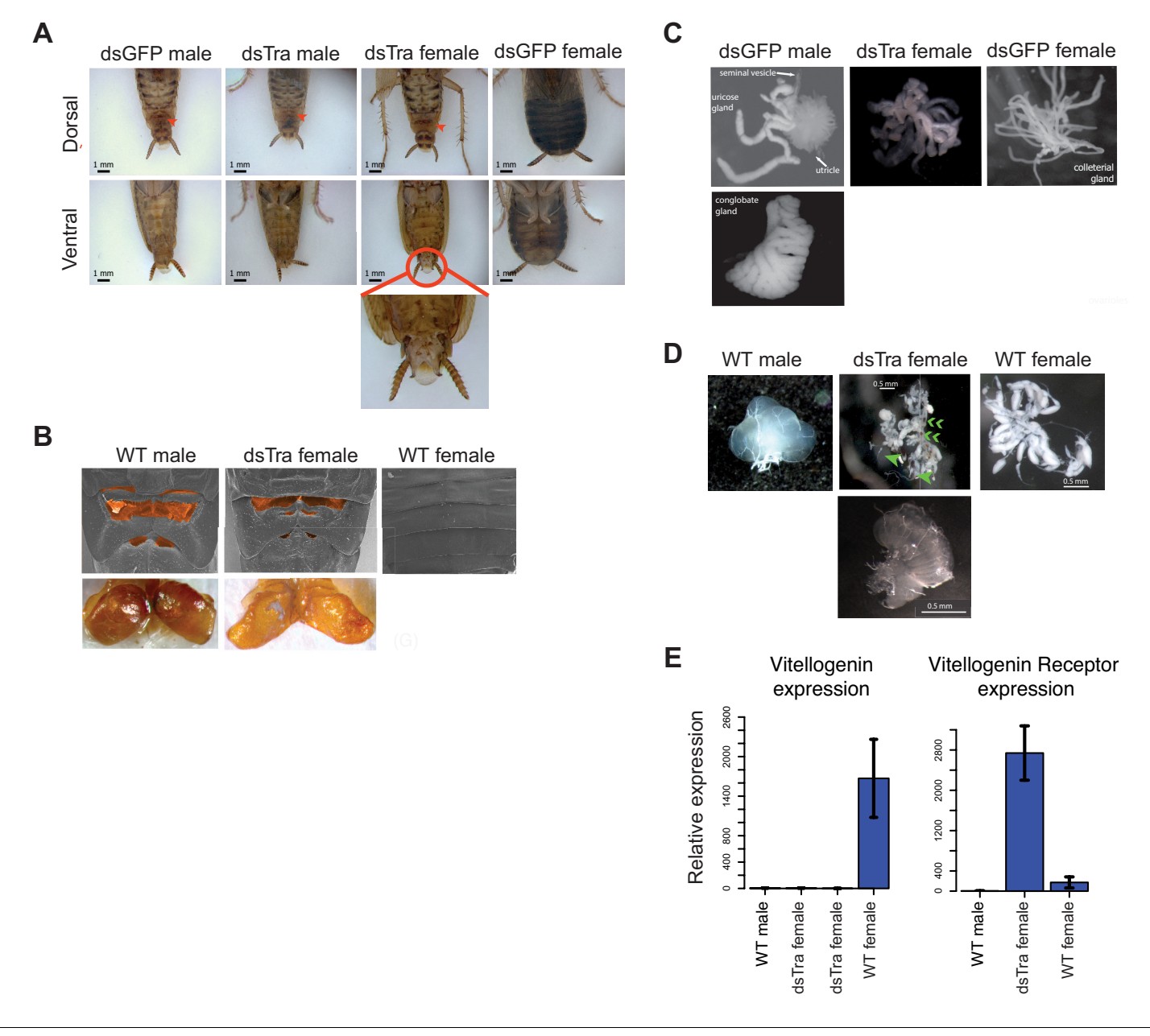

**Figure 3.** *BgTra* is necessary for female-specific but not male-specific sexual differentiation in *Blattella germanica*. (**A**) *dsBgTra* females, injected with *dsBgTra* in their 4th, 5th, and 6th instars, molt to adults with a masculinized abdomen, in contrast to control animals injected with dsGFP. Top row, dorsal view. Bottom row, ventral view. The tergal gland in wild-type males and *dsBgTra* females is indicated with red arrows. Red circle outlines abnormal tissue extrusion typical of *dsBgTra* females; zoomed-in view of this tissue is shown below. (**B**) Scanning electron microscopy (top row) and light microscopy (bottom row) reveal developed male-like tergal glands (shaded orange in SEMs and dissected for light microscopy) in *dsBgTra* females compared to wild-type (WT) females. (**C**) *dsBgTra* females have malformed, partly masculinized colleterial glands. Left column shows wild-type male accessory glands composed of seminal vesicles, utricles, uricose glands, and a conglobate gland. Right column, wild-type female colleterial gland. Center column, gland dissected from *dsBgTra* female is located in the same position as the wild-type colleterial gland, but shows thickened tubules reminiscent of the male accessory glands. (**D**) *dsBgTra* female gonads are deformed or intermediate between male and female gonads. Left column shows testis of a 5-day-old wild-type male. Middle column, top image shows disorganized gonad tissue in a typical 2-week-old *dsBgTra* female. Most individuals (n = 12) had underdeveloped ovaries with some undifferentiated tissue, some ovarioles (green arrows), and some sclerotized cuticle (green chevrons) not normally found in the ovary. Middle column, bottom shows that some *dsBgTra* females (n = 3) have intersex gonads composed of both ovarian and testicular tissue (indicated by green arrow). Right column shows ovaries of a 2-day-old wild-type female. (**E**) *vitellogenin*, an egg yolk protein, production is under the control of *BgTra*. Left: qPCR measuring relative vitellogenin expression in the fat body of males and females treated

*Figure 3 continued on next page*

*Figure 3 continued*

with *dsBgTra*. Right: relative upregulation of vitellogenin receptor transcription in the gonads of *dsBgTra* females, presumably due to the lack of circulating vitellogenin. *vitellogenin* and vitellogenin receptor gene expression was normalized to actin.

DOI: https://doi.org/10.7554/eLife.47490.009

The following figure supplements are available for figure 3:

**Figure supplement 1.** *dsBgTra* reduces *BgTra* expression in males and females of *Blattella germanica*.
DOI: https://doi.org/10.7554/eLife.47490.010

**Figure supplement 2.** Abnormal morphology in 6th instar *dsBgTra* female nymphs of *Blattella germanica*.
DOI: https://doi.org/10.7554/eLife.47490.011

production, was also partly masculinized, displaying thickened tubules reminiscent of male accessory glands (*Figure 3C*).

In contrast, male nymphs subjected to an equivalent *BgTra* RNAi treatment progressed normally through development. All *dsBgTra* male nymphs molted into adult males with wild-type external and internal morphology (n = 20); these adults were fertile, producing broods of normal size and number.

In *D. melanogaster*, the sex determination pathway regulates the production of sex-specific cuticular hydrocarbons (CHCs) (*Ferveur et al., 1997*; *Shirangi et al., 2009*). We find that in *B. germanica*, male and female CHC production is controlled by *BgTra*. The CHC profile of *dsBgTra* females is more similar to that of wild-type males than wild-type females; in particular, the precursor of the female contact sex pheromone is depleted in *dsBgTra* females, while the abundance of male-biased CHCs is increased (*Figure 4—figure supplement 2*). *dsBgTra* females have significantly more total CHCs than either wild-type males or wild-type females (*Figure 4—figure supplement 2*), a phenomenon likely explained by their non-functional ovaries. In wild-type females, large amounts of hydrocarbons are provisioned to the maturing oocytes (*Fan et al., 2008*). In ovariectomized wild-type females, some of the excess hydrocarbons which were destined to provision the ovaries are shunted to the cuticle (*Schal et al., 1994*); a similar process may be at work in *dsBgTra* females.

In addition to its female-specific effect on external and internal morphology, *BgTra* RNAi had a female-specific effect on gene expression. Expression of vitellogenin (yolk protein) in the fat body is high in wild-type adult females and absent in males. In *dsBgTra* females, vitellogenin expression was reduced to undetectable levels, similar to wild-type males (*Figure 3E*). Perhaps in response to the shortage of vitellogenin, expression of the vitellogenin receptor was upregulated in the gonads of *dsBgTra* females (*Figure 3E*). Overall, we conclude that *BgTra*, like its holometabolous insect orthologs, is required for female sexual differentiation but is dispensable in males. This is despite the fact that, in contrast to holometabolous insects, *BgTra* produces functional isoforms in males as well as in females.

## *tra* represses male-specific behavior in female cockroaches

In *D. melanogaster*, sex-specific behavior is indirectly under the control of *tra* via the *fruitless* (*fru*) transcription factor (*Ryner et al., 1996*; *Heinrichs et al., 1998*; *Kimura et al., 2005*). *fru* orthologs also control courtship behavior in other dipterans and in *B. germanica*, and *fru* shows conserved sex-specific splicing across holometabolous insects (*Meier et al., 2013*; *Clynen et al., 2011*; *Salvemini et al., 2010*). In *D. melanogaster* females, Tra prevents expression of *fru* isoforms that direct male courtship behavior (*Heinrichs et al., 1998*). To test whether *BgTra* also controls sex-specific behavior in *B. germanica*, we exposed ten masculinized *dsBgTra* female adults to antennae clipped from sexually mature wild-type females. In typical courtship, wild-type males are stimulated by a contact sex pheromone on the female antenna. Upon contact with a female antenna, wild-type males orient their abdomen towards the antenna and raise their wings to display the tergal glands. As the female feeds on tergal gland secretions, the male extends his abdomen, grasps the female's genitalia, and copulation follows (*Roth and Willis, 1952*). Wild-type females do not show the wing-raising behavior. Nine out of ten masculinized *dsBgTra* female adults showed the same response as wild-type males, although the mean lag time between introduction of the female antenna and wing raising was longer in *dsBgTra* females (mean = 19.9 s) than in wild-type males (mean = 4.9 s) (Welch's t-test, p=0.02) (*Figure 4A,B*). These results indicate that *dsBgTra* females, unlike wild-type

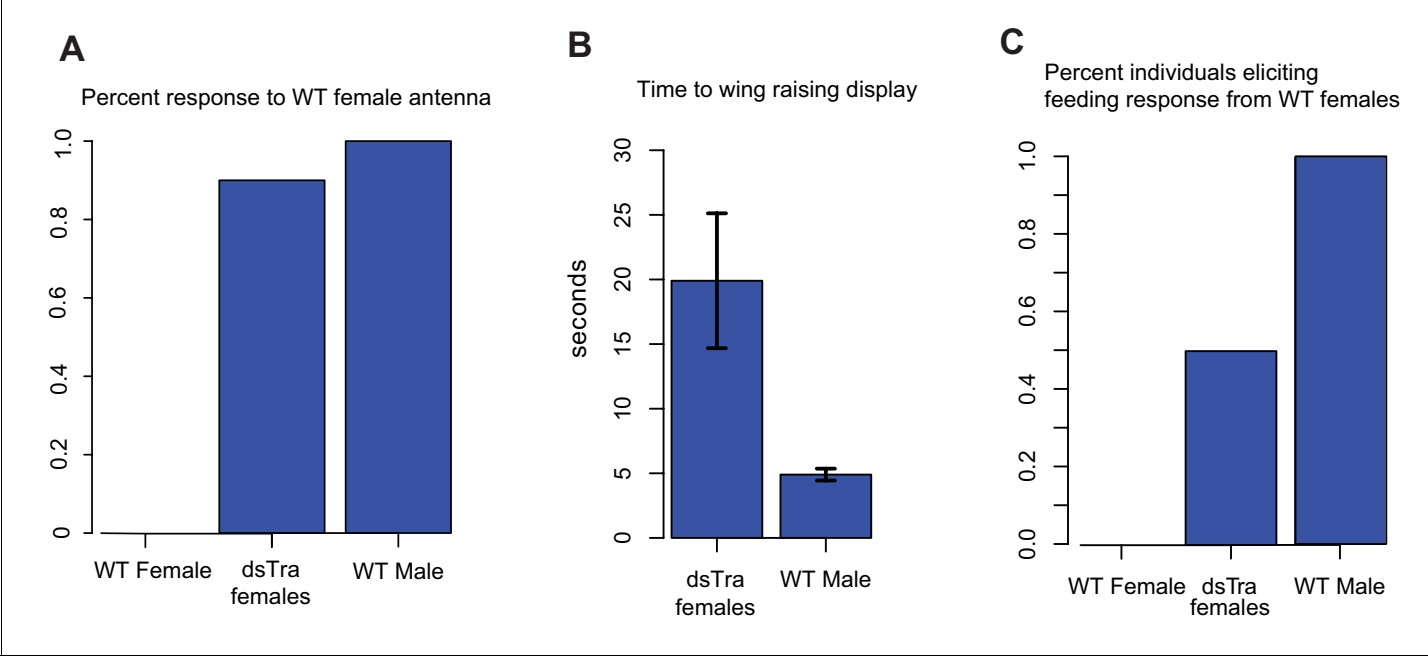

**Figure 4.** *dsBgTra* females perform male courtship behavior and elicit courtship responses from wild-type females. In courtship, wild type male *Blattella germanica* raise their wings to display their tergal gland. When courtship is successful, wild type females feed upon the secretions of this gland prior to copulation. *dsBgTra* females (n = 10), wild-type females (n = 10), and wild-type males (n = 10) were exposed to an antenna clipped from a wild-type, 7 day old virgin female, and their response times in seconds were compared. (**A**) Nine out of ten *dsBgTra* females performed the stereotypical male wing-raising courtship display in response to a female antenna compared to 0/10 of wild-type females within the one minute of observation time. (**B**) However, *dsBgTra* females took longer than wild-type males to respond with the wing-raising display. Wild-type females did not respond to the stimulus throughout the duration of the one-minute observation time. (**C**) 5/10 *dsBgTra* females elicited feeding response from wild-type females after raising their wings. Wild-type females, lacking tergal glands, do not elicit a feeding response from other wild-type females.
DOI: https://doi.org/10.7554/eLife.47490.012

The following figure supplements are available for figure 4:

**Figure supplement 1.** *BgTra* controls sex-specific oligosaccharide synthesis in tergal glands (**A**) Relative abundance of oligosaccharides in tergal glands derived from *dsBgTra* females (dsTra1-dsTra10) and wild-type males (wt1-wt10).
DOI: https://doi.org/10.7554/eLife.47490.013

**Figure supplement 2.** *BgTra* controls production of sex-specific cuticular hydrocarbons.
DOI: https://doi.org/10.7554/eLife.47490.014

females, perceive wild-type female sex pheromone and respond with courtship behavior. Similar to its function in holometabolous insects, *BgTra* is required to repress male-specific behavior in *B. germanica*.

In a separate set of experiments, we tested whether wild-type females respond to the wing-raising display of masculinized *dsBgTra* females. In five out of ten instances, wild-type females began feeding on the tergal glands of *dsBgTra* females, confirming the functionality of these glands (*Figure 4C*). Overall, the results indicate that *dsBgTra* females express male sexual traits that attract wild-type females.

### *BgTra* is necessary for female, but not male embryonic viability

We also found that *BgTra* is crucial for female embryonic development. Three-day-old adult females injected with *dsBgTra* produced all-male broods (*Figure 5A*), suggesting that maternal *dsBgTra* treatment either masculinizes or kills female progeny. Female-specific lethality appears more likely for two reasons. First, the number of offspring in the all-male broods from *dsBgTra* females was about 50% of the offspring from dsGFP-injected control females (*Figure 5B*). Second, all the broods that hatched from *dsBgTra* mothers contained dead embryos remaining in the ootheca (*Figure 5C*). One possibility is that *BgTra* may affect dosage compensation in the cockroach, which has male, XX/XO sex determination (*Stevens, 1905*). When *T. castaneum* females are injected with dsRNA

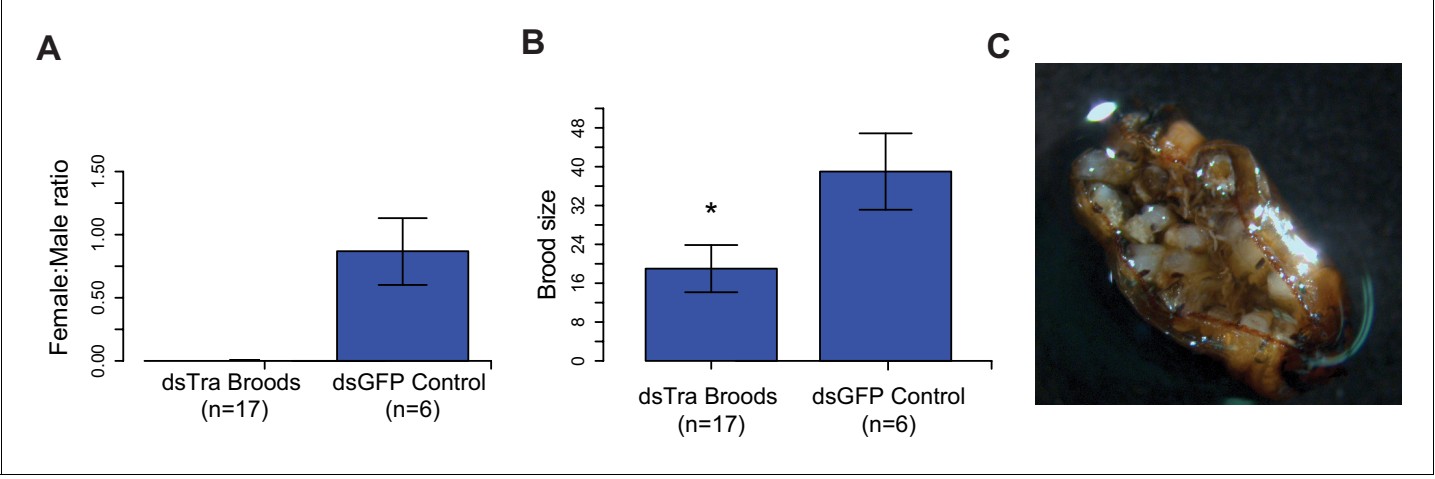

**Figure 5.** Maternal knockdown of *BgTra* in *Blattella germanica* results in all-male broods.. (A) Females injected with *dsBgTra* (n = 17) three days after emergence and mated to wild type males produced all-male broods, compared to control females injected with dsBgGFP (n = 6) (B) Brood sizes from mothers injected with *dsBgTra* were significantly smaller than the dsGFP controls (Welch's t-test, p=0.001). (C) Image of a typical ootheca (egg case) from a *dsBgTra* mother that failed to hatch. Of 22 mothers injected with dsTra, five egg cases failed to hatch completely. The remaining oothecae had 5–19 dead embryos remaining within it after all males had emerged. Variability in the number of dead embryos can be attributed to surviving offspring eating dead embryos, which was observed by the authors.

DOI: https://doi.org/10.7554/eLife.47490.015

targeting *tra*, they also produce all-male broods with much reduced survival (*Shukla and Palli, 2012a*). These results suggest that *tra* could either have a deeply conserved role in dosage compensation, or has repeatedly evolved a role in this process.

## *B. germanica BgTra* controls sex-specific splicing of *BgDsx*

In *B. germanica*, *BgDsx* has two female-specific and one male-specific isoform (*Figure 2C*). Because *dsx* splicing is under the control of *tra* in most holometabolous insects, we tested whether *BgTra* RNAi affected the production of sex-specific *BgDsx* isoforms. qPCR and RT-PCR revealed that in *dsBgTra* females, *BgDsx* splicing switched completely from the female to the male pattern (*Figure 6*). Neither of the two female-specific transcripts were present in *dsBgTra* females, whereas the male-specific *BgDsx* isoform was present in abundance similar to wild-type males (*Figure 6*). As expected, no change in *BgDsx* splicing was observed in *dsBgTra* males (data not shown). This is similar to the pattern of *tra*-dependent female *dsx* splicing seen in holometabolous insects such as *D. melanogaster* and *T. castaneum* (*Shukla and Palli, 2012b*; *Inoue et al., 1992*). However, in contrast to holometabolous insects, male-specific *dsx* splicing in the cockroach is unlikely to be due to the absence of functional Tra protein in males, since *BgTra* produces the same full-length transcript isoforms in both sexes (*Figure 1D–F*).

## *BgDsx* is necessary for male-specific but not female-specific sexual differentiation

In separate experiments, we injected three different dsRNAs targeting *BgDsx*. Two of these dsRNAs targeted the DNA-binding domain shared by male and female *BgDsx* transcripts; a third dsRNA targeted the Dsx Dimerization Domain, also shared between males and females (*Figure 2*). Surprisingly, we observed that *BgDsx* transcript abundance increased after injection with *dsBgDsx* in both males and females (*Figure 7—figure supplement 1*), making it difficult to measure the efficiency of RNAi treatment. Increased expression of a target gene after dsRNA treatment has been previously described (*Nakamura and Extavour, 2016*), and there is not always a strong relationship between transcript abundance and RNAi phenotype (*Xiang et al., 2017*). Most plausibly, the upregulation observed after a dsRNA treatment results from a rebound effect on the transcription of the targeted gene. Although the effect of RNAi on *BgDsx* transcript abundance was similar in males and females,

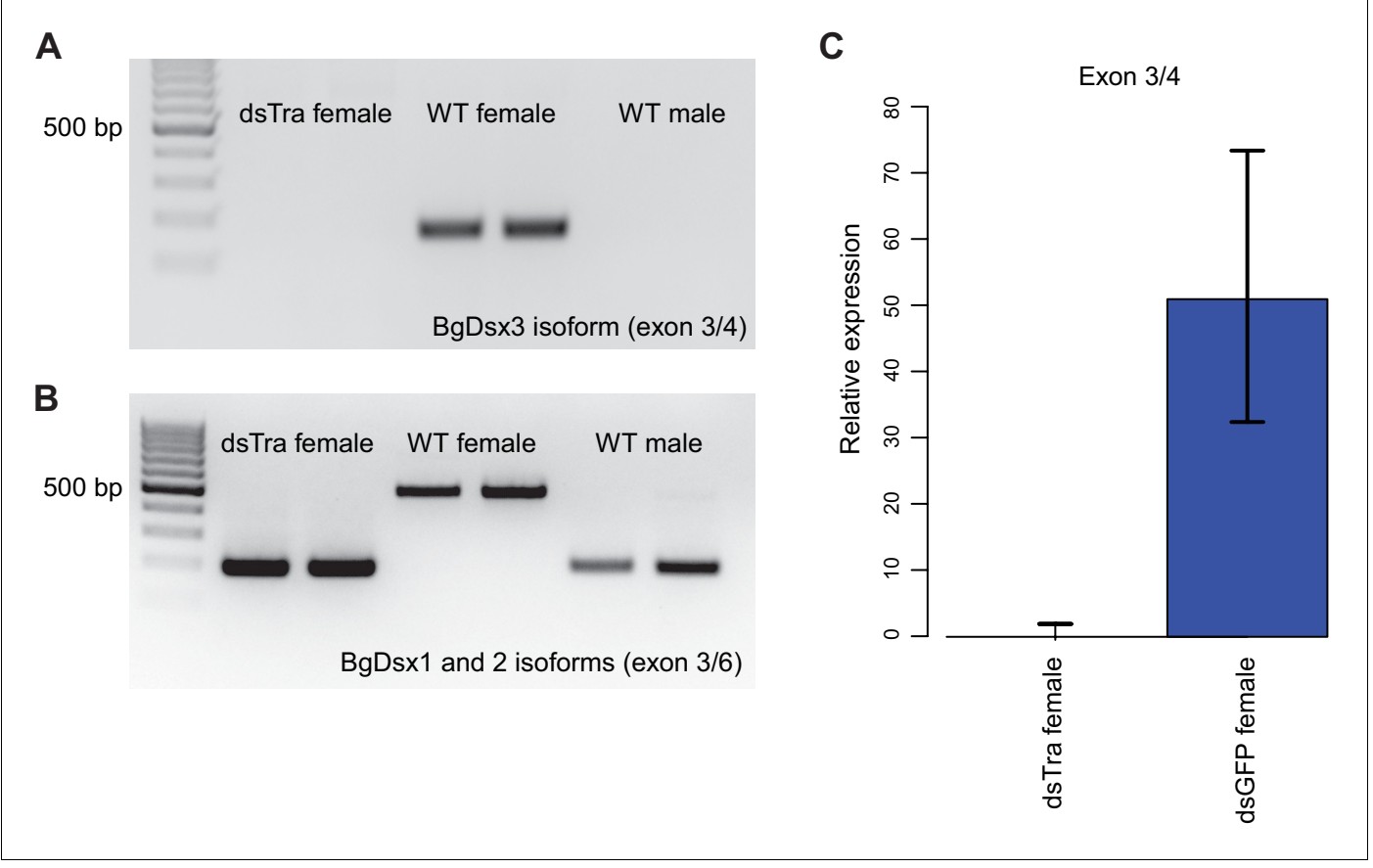

**Figure 6.** *BgTra* controls sex-specific splicing of *BgDsx*. (**A**) RT-PCR showing that the female-specific *BgDsx* exon 3/4 junction (*Figure 2B,C*) is absent in *dsBgTra* females (**B**) RT-PCR showing that the male-specific *BgDsx* exon 3/5 junction (*Figure 2B*) is present in *dsBgTra* females. *dsBgTra* females do not express any of the female-specific *BgDsx* isoforms. (**C**) qPCR shows complete absence of the female-specific *BgDsx* exon 3/4 junction in *dsBgTra* females.

DOI: https://doi.org/10.7554/eLife.47490.016

its effect on adult morphology and downstream gene expression was markedly different between the two sexes.

Both males and females injected with *dsBgDsx* during 4th, 5th, and 6th instars progressed normally through nymphal development. However, upon the adult molt, all males injected with *dsBgDsx* targeting the DM domain (n = 23) had an abnormal extrusion of tissue at the tip of their abdomen (*Figure 7A*). This extrusion appears to be an outgrowth of the ejaculatory duct (*Figure 7—figure supplement 2*). A range of body shapes was observed in these adults, from slender abdomens typical of wild-type males to the rounder abdomens typical of females. The tergal glands of *dsBgDsx* males also ranged from fully developed to severely reduced (*Figure 7A*). Moreover, the *dsBgDsx* males showed darker pigmentation, similar to wild-type females, especially on the dorsal side (*Figure 7A*). Because of their malformed ejaculatory ducts, *dsBgDsx* males were not able to mate, making it impossible to assess their fertility. Dissection of *dsBgDsx* males revealed apparently normal testes (n = 15), with morphologically normal sperm indistinguishable from wild-type *B. germanica* sperm (n = 3). However, the conglobate glands of *dsBgDsx* males failed to mature over the first week of adult life, while tissue discoloration was observed in the utricles (*Figure 7—figure supplement 3*).

Females injected with BgDsDsx targeting either the DM domain or the Dsx Dimerization domain molted into adults with no visible external abnormalities (*Figure 7A*), and had apparently normal gonads and reproductive tracts. Adult *dsBgDsx* females elicited courtship from wild-type males, mated with them, and produced broods whose size was not significantly different from those of

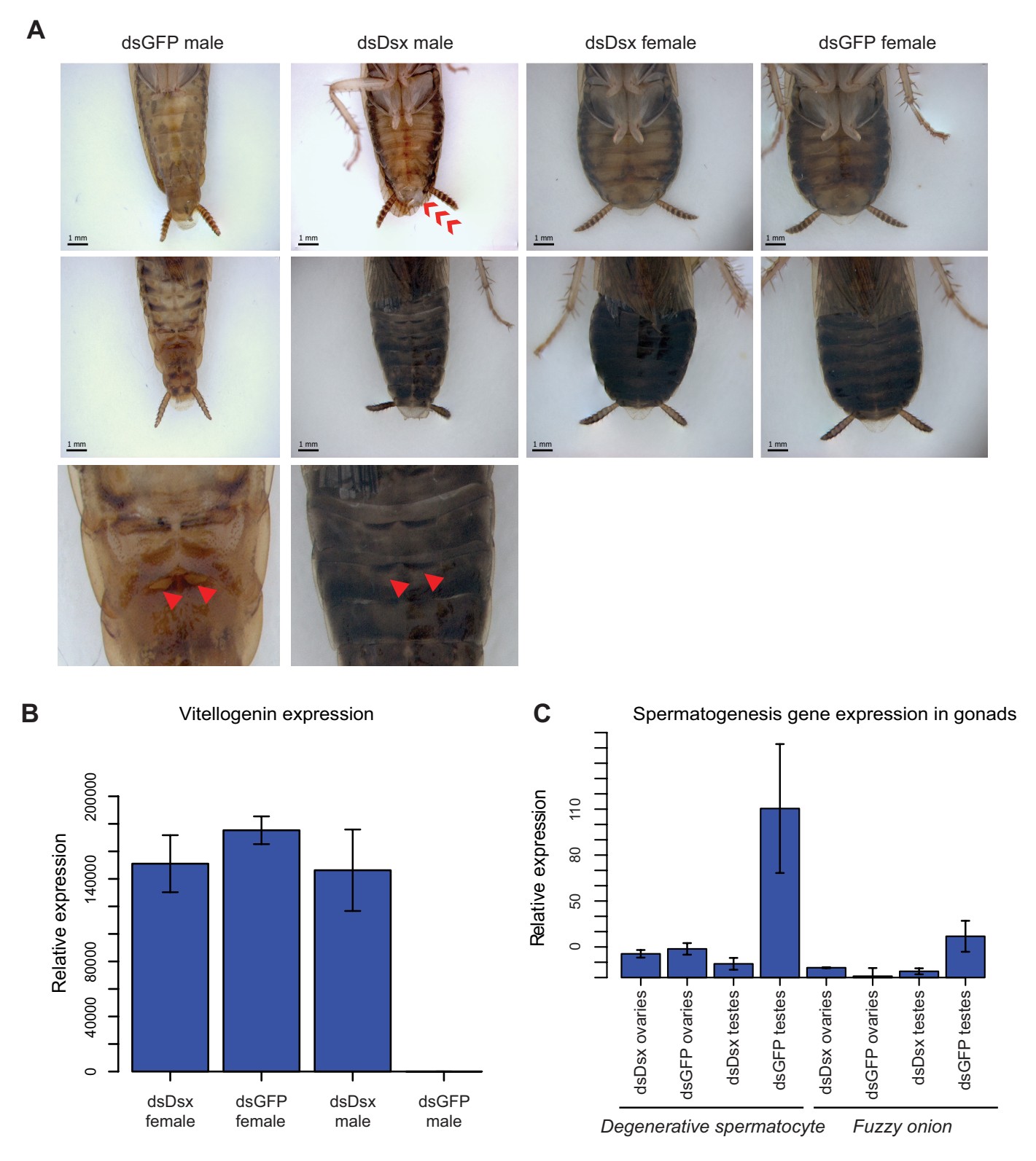

**Figure 7.** *dsBgDsx* RNAi in *Blattella germanica* feminizes males but has no effect in females (**A**) *dsBgDsx* males have reduced tergal glands (red arrow), darker female-like pigmentation, and an extrusion of tissue at the distal end of the abdomen (red chevron) compared to dsGFP injected males (see also *Figure 7—figure supplement 3*). Females are unaffected by *dsBgDsx* treatment. Bottom row shows zoomed-in view of dorsal segments containing the tergal gland. The openings from which tergal glands secrete their oligosaccharides and lipids in dsGFP males (red arrows) are greatly reduced in
*Figure 7 continued on next page*

*Figure 7 continued*

*dsBgDsx* males. (B) *BgDsx* represses *vitellogenin* expression in males, but has no effect in females (n = 3 biological replicates per treatment). (C) *BgDsx* controls the expression of two spermatogenesis-related genes in males, but has no effect on their expression in females. qPCR showing the expression of two genes important for *Drosophila melanogaster* spermatogenesis, *degenerative spermatocyte* and *fuzzy onions*, relative to *actin* (n = 3 biological replicates for all columns).

DOI: https://doi.org/10.7554/eLife.47490.017

The following figure supplements are available for figure 7:

**Figure supplement 1.** *BgDsx* RNAi causes an increase in *BgDsx* transcript abundance in both males and females of *Blattella germanica*.

DOI: https://doi.org/10.7554/eLife.47490.018

**Figure supplement 2.** Representative sampling of tissue extrusions found in *dsBgDsx* males of *Blattella germanica* (light blue arrowheads).

DOI: https://doi.org/10.7554/eLife.47490.019

**Figure supplement 3.** The conglobate glands and utricles of *dsBgDsx* males of *Blattella germanica* fail to properly mature.

DOI: https://doi.org/10.7554/eLife.47490.020

**Figure supplement 4.** *BgDsx* has no effect on brood size in females of *Blattella germanica* that were injected with *dsBgDsx* either at multiple stages throughout nymphal development (A) or as virgin adult females (B).

DOI: https://doi.org/10.7554/eLife.47490.021

control females (*Figure 7—figure supplement 4*). When virgin females received *dsBgDsx* injections 3 days before mating, they produced broods of normal size with normal sex ratio (*Figure 7—figure supplement 4*). Thus, it appears that *BgDsx* controls many though not all male-specific traits, but has no obvious effect on female-specific development. This is clearly different from holometabolous insects, where *dsx* plays active roles in both male and female sexual differentiation, and where *dsx* mutants develop into intersexes that are intermediate in phenotype between males and females (*Baker and Ridge, 1980*).

To test whether *BgDsx* controlled sex-specific gene expression, we examined the expression of two genes, *fuzzy onions* (*fzo*) and *degenerative spermatocyte* (*des*), that function in spermatogenesis in *D. melanogaster* and have homologs throughout metazoans (*Hales and Fuller, 1997*; *Mozdy and Shaw, 2003*; *Endo et al., 1996*; *Endo et al., 1997*). Expression of both genes was greatly reduced in the testes of *dsBgDsx* males compared to the testes of wild-type males, whereas no significant effect was seen in the ovaries of *dsBgDsx* females (*Figure 7C*). Conversely, the female-specific *vitellogenin* gene was strongly upregulated in the fat body of *dsBgDsx* males compared to wild-type males, reaching the level normally seen in wild-type females (*Figure 7B*). In contrast, vitellogenin expression was not affected in *dsBgDsx* females (*Figure 7B*). Together, these results suggest that *BgDsx* controls downstream gene expression in males but not in females. This pattern, while different from holometabolous insects, has been observed in the hemipteran *N. lugens*, where a *dsx* ortholog has a female-specific isoform that appears to play no role in vitellogenin production (*Zhuo et al., 2018*). In *D. melanogaster*, yolk protein genes are upregulated by the female-specific isoform of *dsx* and repressed by the male-specific isoform, so that *dsx* mutants show an intermediate level of yolk protein gene expression (*Coschigano and Wensink, 1993*).

## Discussion

The sex determination pathway must by necessity produce a bimodal output: male or female. How this output is achieved varies considerably across animal taxa and can rely on molecular processes as different as systemic nuclear hormone signaling, post-translational protein modification, or, in the case of insects, sex-specific alternative splicing. In this paper, we examined sexual differentiation in hemimetabolous insects to investigate the evolutionary origin of this unique mode of sexual differentiation.

The canonical *transformer-doublesex* pathway of holometabolous insects has three derived features that are not found in non-insect arthropods or in non-arthropod animals: (*Charlesworth, 1996*) the presence of distinct male- and female-specific splicing isoforms of *dsx* that play active roles in both male and female sexual development; (*Graves, 2006*) the production of functional Tra protein in females but not males; and (*Ellegren, 2000*) the control of female-specific *dsx* splicing by Tra. Our work in hemimetabolous insects suggests that these elements evolved separately and at different times, and that the definitive *tra-dsx* axis was assembled in a stepwise or mosaic fashion.

## Sex-specific splicing of *doublesex* predates its role in female sexual differentiation

Distinct male and female *dsx* isoforms have been reported in all holometabolous insects examined, as well as in the planthopper *N. lugens* (Hemiptera) (*Zhuo et al., 2018*). Our results confirm the presence of male and female *dsx* isoforms in another hemipteran, *R. prolixus,* and show that this pattern is conserved through Blattodea – the most basal insect order studied to date. This suggests that sex-specific splicing of *dsx* was likely the first feature of the canonical insect sexual differentiation pathway to evolve. In *P. humanus* (Phthiraptera), the alternative splicing of *dsx* follows the usual pattern, with a common N-terminal domain and alternative C-termini, but we found that both *dsx* isoforms are present in both sexes. Previous work in the hemipteran *Bemisia tabaci* also failed to detect sex-specific *dsx* isoforms (*Guo et al., 2018*), suggesting that some insects may have secondarily lost sex-specific *dsx* splicing.

We did not detect any role for the cockroach *dsx* gene in female sexual differentiation in *B. germanica*. Recent work in the Hemipteran *N. lugens* produced similar results: despite the presence of male and female *dsx* isoforms, *N. lugens* females developed normally following *dsx* RNAi knockdown, while the males were strongly feminized (*Zhuo et al., 2018*). Together, these results suggest that in at least some hemimetabolous insects, *dsx* is spliced as in holometabolous insects, but may function as in the crustacean *D. magna*, where *dsx* is necessary for male but not female development (*Kato et al., 2011*).

The existence of a female-specific *dsx* isoform in the absence of a female-specific *dsx* function is, of course, surprising. Although we examined multiple features of female external and internal anatomy, gene expression, behavior, and reproduction in *B. germanica dsBgDsx* females, we cannot rule out that *dsx* performs some subtle or spatially restricted function in female cockroaches. Such relatively minor and tissue-specific role, if it exists, could provide a crucial stepping stone in the origin of the classical *dsx* function as a bifunctional switch that is indispensable for female as well as male development. One possibility is that, similar to other transcription factors, *dsx* originally evolved alternative splicing as a way of promoting the development of different cell types. If some *dsx* isoforms evolved functions that were dispensable for male sexual development, *dsx* could gradually lose its ancestral pattern of male-limited transcription, opening the way for the evolution of new isoforms with first subtle, but eventually essential, functions in females.

## The role of *tra* in sexual differentiation predates canonical sex-specific splicing of *tra*

The opposite pattern is observed for transformer, which apparently evolved a sex-specific role before that role came to be mediated by a male-specific stop codon. In most holometabolous insects, alternative splicing of *tra* produces a functional protein in females but not in males, due to a premature stop codon in a male-specific exon near the 5' end of *tra*. In hemimetabolous insects, we only observe this type of sex-specific splicing in *R. prolixus*. In *B. germanica* and *P. humanus*, full-length and presumably functional Tra proteins that include all of the important functional domains are produced in both males and females. The female-limited function of Tra in the cockroach could result from higher expression of truncated *BgTra* isoforms in females (see below), or from sex-specific expression of unknown cofactors of Tra. In either case, the function of *tra* in female sexual differentiation predates the evolution of female-limited Tra protein expression.

Although they lack the canonical sex-specific splicing observed in holometabolous insects, the *tra* genes of *B. germanica* and *P. humanus* show a different pattern of alternative splicing affecting the CAM domain, which plays a role in *tra* autoregulation in holometabolous insects (*Tanaka et al., 2018*). In some *tra* isoforms, read-through of an exon containing the 5' half of the CAM domain results in a premature stop codon; in *B. germanica* and *P. humanus*, these transcripts are found in both sexes. In *BgTra*, isoforms with an intact reading frame come in two types. Some of these intact isoforms contain an in-frame exon spliced into the middle of the CAM domain; in the others, this exon is spliced several base pairs upstream of the stop codon, producing an uninterrupted CAM domain. In *P. humanus*, we only isolated *tra* isoforms with an exon interrupting the CAM domain. The splice junction in the middle of the CAM domain is conserved in holometabolous insects (*Hediger et al., 2010*), and the amino acid residues flanking this junction are necessary for female-specific autoregulation of *tra* in the housefly (*Tanaka et al., 2018*). It remains to be determined

whether the CAM domain is involved in *tra* autoregulation in hemimetabolous insects, but if so, the truncated or interrupted isoforms would likely be incapable of autoregulation, with potentially important consequences for the splicing of *tra* and its downstream targets. Interestingly, the evolution of the canonical holometabolous splicing of *tra*, with a male-specific premature stop codon, coincided with a loss of *tra* isoforms with an interrupted CAM domain. In fact, the splicing pattern of *tra* in *R. prolixus* suggests that the male-specific stop codon originally evolved in the middle of the CAM domain (*Figure 1A*). This change may indicate a transition from an ancestral state where both sexes had functional as well as non-functional Tra isoforms, to the derived holometabolous condition where functional Tra protein is confined to females, and non-functional Tra to males.

## The role of *tra* in regulating *dsx* splicing predates sex-specific expression of Tra protein

Although the function of *tra* in female sexual development in *B. germanica* does not appear to be mediated by *dsx*, we find that *tra* is necessary for sex-specific *dsx* splicing in the cockroach, as it is in all holometabolous insects except Lepidoptera (*Kiuchi et al., 2014*). The mechanism of *dsx* regulation by *tra* is likely to be different in *B. germanica* compared to holometabolous insects. In the latter, functional Tra protein is absent in males but present in females, where it dimerizes with the RNA-binding protein Tra-2 to regulate the splicing of *dsx* and *fru* (*Hoshijima et al., 1991*; *Heinrichs et al., 1998*; *Nagoshi et al., 1988*). In the cockroach, however, full-length Tra proteins containing all the functional domains are expressed at similar levels in both sexes, so that male-specific splicing of *dsx* in males cannot be explained by lack of Tra. It could be due instead to an interaction of Tra with different binding partners in males vs females. We note that while the *D. melanogaster* Tra protein does not contain an RNA-binding domain and does not bind to RNA without Tra-2, the cockroach Tra protein, as well as those of some other insects and crustaceans, contain predicted RNA recognition motifs (*Inoue et al., 1992*). Thus, it is possible that Tra first evolved its function in regulating alternative splicing as a broadly acting RNA-binding protein that worked in concert with other splicing factors, before losing its ability to bind RNA and becoming a dedicated partner of Tra-2 with a narrow range of downstream targets. If so, the situation we find in *B. germanica* may represent a transitional stage in the evolution of the canonical *tra-dsx* axis, where *dsx* is one of many Tra targets, rather than the main mediator of its female-specific function. In this scenario, the *tra-dsx* axis evolved via merger between expanding *dsx* function (from males to both sexes) and narrowing *tra* function (from a general splicing factor to the dedicated regulator of *dsx*).

## Evolution of the insect sexual differentiation pathway: stepwise or mosaic?

A more extensive sampling of hemimetabolous insects will no doubt be necessary to reconstruct the full series of events that produced the familiar holometabolous mode of sexual differentiation, and to determine the timing of the key synapomorphies among the variety of lineage-specific changes. Based on our limited taxon sampling, we can propose a tentative model. We suggest that the canonical insect mechanism of sexual differentiation based on sex-specific splicing of *dsx* and *tra* evolved gradually in hemimetabolous insects, and may not have been fully assembled until the last common ancestor of the Holometabola (*Figure 8*). In crustaceans, mites, and non-arthropod animals, *dsx* and its homologs act as male-determining genes that are transcribed in a male-specific fashion and are dispensable for female sexual differentiation (*Kato et al., 2011*; *Li et al., 2018*; *Pomerantz and Hoy, 2015*). One of the earliest events in insects was the evolution of sex-specific *dsx* splicing, where *dsx* is transcribed in both sexes but produces alternative isoforms in males vs females. We do not know whether Tra was involved in sex-specific *dsx* splicing from the beginning, or evolved this function later; in either case, the function of *tra* in controlling female sexual differentiation may well predate its role in regulating *dsx* splicing. Eventually, however, Tra became essential for the expression of female-specific *dsx* isoforms. At the next step, expression of functional Tra protein became restricted to females due to the origin of a male-specific exon with a premature stop codon at the 5' end of the *tra* gene. As this process unfolded, female-specific *dsx* isoforms gradually evolved female-specific functions, which may have been minor at first but became essential at or before the origin of holometabolous insects (*Figure 8*).

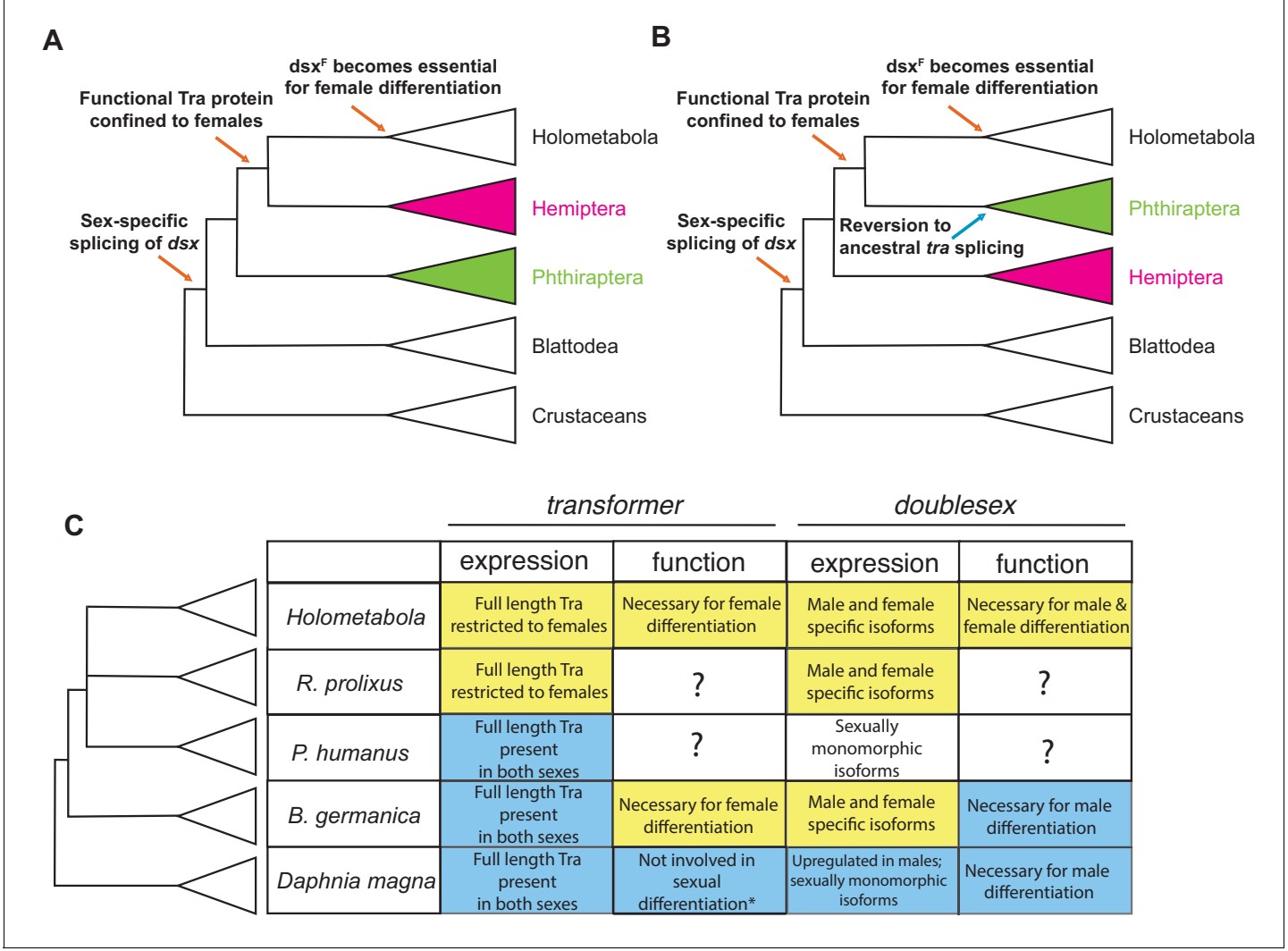

**Figure 8.** Evolutionary assembly of the insect sexual differentiation pathway. Uncertainty in the relationships among Hemiptera, Phthiraptera, and Holometabola (*Misof et al., 2014*; *Johnson et al., 2018*; *Freitas et al., 2018*; *Li et al., 2015*) means that the sequence of events in the evolution of the *tra-dsx* axis is unclear. If Hemiptera (pink) are the sister group of Holometabola, the key molecular innovations have evolved in a stepwise order (A). If Phthiraptera (green) are the sister of Holometabola instead, the evolution of the *tra-dsx* axis must have followed a more mosaic pattern, with at least some secondary reversions (B). (C) Summary of *tra* and *dsx* expression and function in the three hemimetabolous insect orders in comparison to the canonical holometabolous mechanism and to *Daphnia*. No effect on sexual differentiation has been reported following *tra* knock-down in *Daphnia* (asterisk) (*Kato et al., 2011*). Hypothesized ancestral states are shown in blue; hypothesized derived states in yellow. Within Holometabola, subsequent events in the evolution of the *tra-dsx* axis include a secondary loss of *tra* in Lepidoptera, loss of the CAM domain of Tra in *Drosophila*, and the evolution of *Sex-lethal* (*Sxl*) as the upstream regulator of *tra* splicing in *Drosophila*.

DOI: https://doi.org/10.7554/eLife.47490.022

The details of this model depend on the phylogenetic relationships of hemimetabolous insect orders and the holometabolous clade, which remain controversial. Despite rapidly growing amounts of data, phylogenetic analyses have had limited success in identifying the closest outgroup to Holometabola. In some phylogenies, that outgroup is Hemiptera (*Figure 8A*) (*Sasaki et al., 2013*); in others it is Psocodea, which includes lice (*Figure 8B*) (*Ishiwata et al., 2011*; *Misof et al., 2014*; *Johnson et al., 2018*); in all molecular phylogenies to date, the relevant nodes are not strongly supported. In fact, some phylogenies show Hemiptera and Psocodea as a polytomy (*Freitas et al., 2018*; *Li et al., 2015*). The holometabolous splicing pattern of *tra* is seen in *R. prolixus* but not in the cockroach or louse, suggesting that either Hemiptera are the closest outgroup to holometabolous insects, or the splicing of *tra* in *P. humanus* represents a reversion to an ancestral state (*Figure 8*).

Our study of *transformer* and *doublesex* expression and function across hemimetabolous insects establishes a broad outline for the origin of the unique insect-specific mode of sexual differentiation via alternative splicing. Comparative and functional work in other basal insects, non-insect hexapods, and crustaceans will be needed to determine the ancestral functions of *tra*, the roles of female-specific *dsx* isoforms in hemimetabolous insects, and other details of this emerging model.

## Materials and methods

### DMRT and SR family gene trees

To differentiate *dsx* orthologs from other DMRT paralogs, we made phylogenetic trees of arthropod DMRT genes. DMRT protein sequences from previously studied arthropods (*Wexler et al., 2014*) (*D. melanogaster, A. mellifera, B. mori, T. castaneum, R. prolixus, D. magna,* and *I. scapularis*) were used as queries (*Supplementary file 4*) in BLASTP searches of predicted gene models from 22 other arthropod species (*Supplementary file 1*). Hits with e-values lower than 0.01 were retained and added to the file that included the original DMRT queries. This sequence file was multiply aligned with the MAFFT program using the –geneafpair setting (*Katoh et al., 2002*). Upon alignment, sequences without the characteristic DMRT DNA-binding domain CCHHCC motif were discarded, and the entire multiple alignment was rerun.

We used a similar procedure to identify *tra* orthologs in hemimetabolous insects. For the RS family gene tree, the query file consisted of Tra, Transformer-2, Pinin, and SFRS protein sequences from *D. melanogaster, B. germanica*, and *D. magna* (*Supplementary file 5*), and a BLASTP search was performed on the same arthropod species, retaining hits with e-values <0.01. After sequences were multiply aligned with the MAFFT algorithm, gaps in the alignment were trimmed with trimAl using the –automated1 setting (*Capella-Gutierrez et al., 2009*).

To construct gene trees for DMRT and RS families, prottest3.4 (*Darriba et al., 2011*) was used for model selection for the amino acid alignment; RAxML8.2.9 was used to build a maximum likelihood tree with a WAG + G model for both the SR family gene tree and the DMRT gene tree (*Stamatakis, 2014*).

### Cockroach husbandry

Functional experiments were conducted with two German cockroach *B. germanica* strains: Orlando Normal, obtained from Coby Schal's laboratory at North Carolina State University; and Barcelona strain from Xavier Belles's laboratory at the Institute of Evolutionary Biology. *B. germanica* were kept in an incubator at 29°C and fed Iams dog food *ad libitum*.

### PacBio isosequencing

Pacific Biosystems (PacBio) isosequencing was performed separately on male and female *B. germanica* tissues. We reasoned that sexually dimorphic tissues were likely to express *doublesex* and *transformer*, including any sex-specific isoforms of these genes that might exist. We extracted RNA from reproductive organs (gonad and colleterial glands) and fat body of a seven day old virgin female and from reproductive organs (gonad, conglobate gland, accessory glands, and seminal vesicles) and fat body of two seven day old males for library preparation. Tissues were pooled in Trizol, and a separate RNA extraction was done for each sex. Clontech SMARTER PCR cDNA synthesis kit and PrimeSTAR GXL DNA polymerase were used for cDNA synthesis; a cDNA SMRTbell kit was then used to add sequencing adapters. The male and female libraries were run on separate SMRT cells on the PacBio Sequel instrument, and the PacBio SMRTLink software was used for downstream data processing. We used tBLASTn to search the high and low quality polished consensus isoforms for *BgTra* and *BgDsx* transcripts. We obtained additional isoforms searching the sets of circular consensus sequences. All splice junctions from *BgTra* and *BgDsx* Isosequencing-generated transcripts were verified with RT-PCR. When the coding sequence of PacBio-generated isoforms disagreed with the *B. germanica* genome, sequence was verified by Sanger sequencing.

### Illumina RNA sequencing

To identify *dsx* and *tra* isoforms in *R. prolixus* and *P. humanus*, we used Trizol to extract RNA according to the manufacturer's instructions. For *R. prolixus*, we extracted RNA separately from male and

female gonad tissue. For *P. humanus*, adults were sexed and pooled before RNA extraction. For both insects, 1 µg of RNA was used to construct each library using the NEBNext kit and Oligos. Libraries were quantified with NEBNext Quant Kit and sequenced using 150 bp paired-end reads on an Illumina HiSeq4000 machine. Poor quality reads were discarded with FastQC (default parameters) (https://www.bioinformatics.babraham.ac.uk/projects/fastqc/) and trimmed with trimmomatic 0.36 using suggested filters (LEADING:3 TRAILING:3 SLIDINGWINDOW:4:15 MINLEN:36) (*Bolger et al., 2014*). Trinity 2.1.1 with default parameters was used for de novo assembly of the male and female transcriptomes (*Haas et al., 2013*).

## 5'/3' RACE

The Clontech 5'/3' SMARTer Rapid Amplification of cDNA Ends (RACE) kit was used to make RACE libraries from male and female *R. prolixus*, *P. humanus*, and *B. germanica* total RNA. All RNA extractions were performed using Invitrogen TRIzol Reagent according to the provided protocol. For *R. prolixus* and *B. germanica*, we combined gonad tissue, reproductive tract tissue, and fat body from a single male or a single female to make the male and female RACE libraries, respectively. *P. humanus* RNA was extracted from pools of between 6–10 whole adult males or females.

For all species, 5' RACE cDNA fragments were amplified using gene-specific primers (*Supplementary file 7*) and kit-provided UPM primers and Clontech Advantage 2 Polymerase. Thermocycling conditions were: 95°C for two mins; 36 cycles of (95°C for 30 s, a primer-specific annealing temperature for 30 s, 68°C for 2.5 min); 95°C for 7 min. 3' RACE cDNA fragments were amplified using single or nested PCRs with gene-specific primers and either Clontech Advantage 2 Polymerase or SeqAmp DNA Polymerase (*Supplementary file 7*). Clontech Advantage 2 PCRs were performed using the above cycling conditions, and SeqAmp DNA Polymerase touchdown thermocycling conditions were: 5 cycles of (94°C for 30 s, 72°C for 1–2 min); 5 cycles of (94°C for 30 s, 68°C for 1–2 min); 35 cycles of (94°C for 30 s, 66°C for 30 s, 72°C for 1–2 min). Nested PCRs used diluted (1:50) outer PCRs as template, a nested gene-specific primer and the UPM short primer, and 25 cycles of (94°C for 30 s, 65°C for 30 s, 72°C for 2 min). All PCR products were gel-purified using Qiagen QIAquick or Macherey-Nagel NucleoSpin gel and PCR clean up kits. PCR products produced by Advantage 2 Polymerase were cloned into the Invitrogen TOPO PCRII vector and SeqAmp DNA Polymerase PCR products were cloned into linearized pRACE vector (provided with the SMARTer RACE kit) with an In-Fusion HD cloning kit. Between 6–10 colonies were Sanger sequenced per PCR reaction with M13F-20 and M13R sequencing primers.

## Reverse-transcription PCR tests of isoform sex-specificity

With *R. prolixus* and *P. humanus*, we found that cDNA synthesized with the Clontech SMARTer 5'/3' RACE kit amplified well in reverse-transcription PCRs (RT-PCR). We synthesized 3' RACE-ready cDNA from two different female and two different male total RNA samples that were not used for 5'/3' RACE. To test isoforms for sex-limited expression, we amplified isoform-specific sequences in female and male samples using 40 cycles of PCR and SeqAmp DNA Polymerase. To test for sex-specificity of isoforms in *B. germanica*, we isolated total RNA from reproductive tracts and fat bodies of male and female adult roaches using Invitrogen TRIzol reagent according the manufacturer's instructions. These RNA samples were isolated from individuals that were not used to construct PacBio or RACE libraries. We then treated these RNA samples with Promega RQ1 DNaseI, also according to manufacturer's instructions, and primed the RNA for cDNA synthesis using a 1:1 mix of oligo dT:random hexamers. We used Invitrogen SuperScriptIII for reverse transcription, and AccuPower Taq from Bioneer for PCR.

## Reverse-transcription qPCR

RNA for qPCR was isolated with Invitrogen TRIzol reagent and then treated with Promega RQ1 DNaseI in accordance with the manufacturer's instructions for both reagents. Reverse transcription for qPCR was performed with Invitrogen SuperScriptIII or SuperScriptIV after RNA was primed with a 1:1 mix of oligo dTs and random hexamers. qPCR was done on non-diluted or diluted cDNA (1/5x) with BioRad Sso Advanced Universal SYBR Mix and on a BioRad CFX96 machine. We quantified expression in 2–3 samples per sex across three technical replicates each (we tested 3 females and three males from *B. germanica*, and 2 females two males from both *R. prolixus*). Each sample was

prepared with three technical replicates, the mean of which was taken for Ct value of each sample. Target gene expression was normalized to β-*actin* expression using the $\Delta$ Ct method (*Schmittgen and Livak, 2008*). A dilution series was used to test primer pairs for efficiency; only primer pairs with calculated efficiency between 90–110% were used.

## Protein organization

BgDsx domains were identified by NCBI domain predictor software: https://www.ncbi.nlm.nih.gov/Structure/cdd/wrpsb.cgi. For BgTra, the CAM domain was identified by visual inspection after a MAFFT alignment of BgTra protein sequences with previously studied Tra proteins in holometabolous insects (*Katoh et al., 2002*). The RS domain was identified by calculating percent of protein sequence containing arginine and serine residues; the RRM domain was identified by NCBI domain predictor software.

## RNAi cloning and synthesis

Template for dsRNA targeting *BgTra* and *BgDsx* were amplified out of *B. germanica* cDNA using Bioneer AccuPower Taq. The following cycling parameters were used: 95°C for 2 min, then 35 x (95°C for 30 s, annealing temperature for 30 s, 72°C for 30 s), then a final extension of 72°C for 3 min. Templates were cloned into TOPO-TA PCRII from Invitrogen. Template for dsGFP was cloned with the same Taq, using the following cycling parameters: 95°C for 1 min, then 35 x (95°C for 30 s, 55°C for 30 s, 72 for 30 s) following by an extension at 72°C for 3 min. Directional colony screens were performed to identify forward and reverse inserts which were combined, so that a single transcription reaction contained both forward and reverse templates. A Thermo Fisher Scientific MEGAscript RNAi kit was used to prepare dsRNA.

## RNAi injections, dissections and imaging

RNAi experiments were conducted in *B. germanica* due to the ease of culturing this insect. In contrast, *R. prolixus* and *P. humanus* are obligate blood feeders, making them difficult to culture outside of specialized biocontainment facilities. We conducted two sets of experiments for each gene in *B. germanica*. First, nymphs were injected throughout their development to assess the function of *BgTra* and *BgDsx* from the mid-point of juvenile development onwards. Second, virgin adult females were injected before mating in a parental RNAi experiment that tested for possible maternal roles of these genes during embryogenesis. Both nymphs and adults were injected in the abdomen between two sternites. For the first set of experiments, we sexed 4th instar *B. germanica* nymphs and injected each individual once in the 4th instar, once in the 5th instar, and once in the 6th instar. Fourth instar nymphs were injected with ~500 ng of dsRNA in 0.5 μL; 5th and 6th instar nymphs were injected with 1 μg of dsRNA in 1 μL. For *BgDsx*, we injected nymphs with three different constructs in separate experiments (n = 8–15 females or males per experiment). Two of the constructs targeted the DM domain, while the third targeted the Dsx dimerization domain (*Figure 2C*). For *BgTra*, we injected male and female nymphs with two different RNAi constructs (*Figure 1D*) in separate experiments (n = 15 per trial). For parental RNAi experiments, we injected 3-day-old virgin females with 1 μg of dsRNA in 1 μL. For *BgDsx*, we injected adult females in two separate trials (n = 8–11), using one of the dsRNA constructs targeting the DM domain. For *BgTra*, we performed three separate experiments (n = 8–17 females per experiment) using two different constructs (*Figure 1D*).

For parental RNAi, we put injected females in individual containers with 2–5 wild-type adult males per container. Adult males were sacrificed after observing that mating had occurred (either by direct observation of mating or by finding spermatophore remains on filter paper). Adult females were sacrificed after their broods hatched. Adult phenotypes from treated insects were examined alongside control individuals that were injected with dsGFP. Dissections of gonads and other reproductive organs were performed in Ringer Solution. For scanning electron microscopy of tergal glands, adult insect abdomens were removed from the thorax and head, dehydrated in 100 percent ethanol, processed by critical point drying, coated with gold, and imaged on a Philips XL30 SEM microscope. Sample size for injections and subsequent phenotypic analysis was dictated by the availability of appropriately staged and sexed insects. Per trial, we injected a minimum of 8 experimental and three control (dsGFP) animals.

## Oligosaccharide chemistry

After dissection, tergal glands were blended with 1 ml of 80% (v/v) EtOH/H$_2$0 and incubated at 4°C overnight to isolate the oligosaccharides. The supernatant was collected and dehydrated. The oligosaccharides were reconstituted in 1 M NaBH$_4$.and incubated at 60°C for 2 hr. Oligosaccharides were further purified as previously described (*Niñonuevo and Lebrilla, 2009*). Structural elucidation employed an Agilent 6520 Q-TOF mass spectrometer paired with a 1200 series HPLC with a chip interface for structural analysis. A capillary pump was used to load and enrich the sample onto a porous graphitized carbon chip, this solvent contained 3% (v/v) ACN/H$_2$O with 0.1% formic acid at a flow rate of 3 µl/min. A nano pump containing a binary solvent system was used for separation with a flow rate of 0.4 µl/min, solvent A contained 3% (v/v) ACN/H$_2$O with 0.1% formic acid and solvent B contained 90% (v/v) ACN/H$_2$O with 0.1% formic acid. A 45 min gradient was performed with the following conditions: 0.00–10.00 min, 2–5% B, 10.00–20.00 min, 7.5% B, 20.00–25.00 min, 10% B, 25.00–30.00 min, 99% B, 30.00–35.00 min, 99% B, 35.00–45.10, 2% B. The instrument was run in positive ion mode. Data analysis was performed with the Agilent MassHunter Qualitative Analysis (B.06) software. For quantitative determination samples were run on an Agilent 6210 TOF mass spectrometer paired with a 1200 series HPLC with a chip interface with the same chromatographic parameters.

Behavioral assays *dsBgTra* females were reared for one week on food pellets (Purina No. 5001 Rodent Diet, PMI Nutrition International) and distilled water in a temperature-regulated walk-in room at 27 ± 1°C, 40–70% relative humidity, and L:D = 12:12 photoperiod (Light, 20:00 – 8:00). A wild-type strain (wild-type, American Cyanamid strain = Orlando Normal, collected in a Florida apartment >60 years ago) was also kept in the same conditions. Newly emerged wild-type males and wild-type females were kept in separate cages to prevent contact. Sexually mature virgin males (20-day-old) and sexually mature virgin females (5–6 days old) were used in behavioral assays. For assays of the wing-raising display in response to isolated female and male antennae, *dsBgTra* females, wild-type females, and wild-type males were acclimated individually for 1 day in glass test tubes (15 cm x 2.5 cm diameter) stoppered with cotton. Observations were carried out at 17:00-19:00 hr (scotophase) in the walk-in incubator room under red fluorescent lights. Each test tube was horizontally placed under an IR-sensitive camera (Everfocus EQ610 Polestar, Taiwan). The antennae of the tested insects were stimulated by contact with either an isolated wild-type female antenna or wild-type male antenna, which was attached to the tip of a glass Pasteur pipette by dental wax. Observation time was 1 min. A single antenna was used for 2–3 tested insects. The percentage of responders was compared by Chi-square test. The latency of wing-raising display was quantified in seconds, from the start of stimulation to the initiation of the display, and compared by t-test (unpaired, $p < 0.05$).

Observations of nuptial feeding were conducted at 12:00-19:00 hr (scotophase). As above, individual tested insects were acclimated in test tubes, but a single wild-type male or wild-type female was introduced into the test tube and wing-raising and tergal gland feeding behaviors were observed for both insects.

## Cuticular hydrocarbon analysis

To compare cuticular hydrocarbons (CHCs) between control and RNAi treatments, individual cockroaches were extracted for 5 min in 200 µL of hexane containing 10 µg of heptacosane (*n*-C26) as an internal standard. Extracts were reduced to 150 µL and 1 µl was injected in splitless mode using a 7683B Agilent autosampler into a DB-5 column (20 m × 0.18 mm internal diameter ×0.18 µm film thickness; J and W Scientific) in an Agilent 7890 series GC (Agilent Technologies) connected to a flame ionization detector with ultrahigh-purity hydrogen as carrier gas (0.75 mL/min constant flow rate). The column was held at 50°C for 1 min, increased to 320°C at 10 °C/min, and held at 320°C for 10 min. For Principal Components Analysis (PCA), we used the percentage of each of 30 peaks previously identified (*Jurenka et al., 1989*). PCA was conducted in JMP (JMP Pro 12, SAS Institute, Inc, Cary, NC). The amount (mass) of each peak was determined relative to the *n*-C26 internal standard.

## Acknowledgements

We thank Dr. Ian Orchard (*R prolixus*), Deanna Fox (*P humanus*), and Dr. John Clark (*P humanus*) for providing us with tissue samples. Megan Meany provided invaluable assistance dissecting and processing *B. germanica* tergal glands for oligosaccharide analysis. We are grateful to Dr. Carlito Lebrilla for assistance with the oligosaccharide analysis. RNA-sequencing was carried out at the DNA Technologies and Expression Analysis Cores at the UC Davis Genome Center, supported by NIH Shared Instrumentation Grant 1S10OD010786-01.

## Additional information

### Funding

| Funder | Grant reference number | Author |
| --- | --- | --- |
| National Science Foundation | 1650042 | Judith Wexler |
| National Science Foundation | 0955517 | Judith Wexler |
| National Institutes of Health | F32GM120893 | Emily Kay Delaney |
| Ministerio de Economía y Competitividad | CGL2012-36251 | Xavier Belles |
| Ministerio de Economía y Competitividad | CGL2015-64727-P | Xavier Belles |
| Generalitat de Catalunya | 2017 SGR 1030 | Xavier Belles |
| National Science Foundation | IOS-557864 | Coby Schal Ayako Wada-Katsumata |
| University of California, Davis | Chancellor's Postdoctoral Fellowship | Emily Kay Delaney |
| National Institutes of Health | 5R35GM122592 | Artyom Kopp |
| University of California, Davis | Population Biology Graduate Group | Judith Wexler |

The funders had no role in study design, data collection and interpretation, or the decision to submit the work for publication.

### Author contributions

Judith Wexler, Conceptualization, Funding acquisition, Investigation, Visualization, Writing—original draft, Project administration, Writing—review and editing; Emily Kay Delaney, Investigation, Visualization, Methodology, Writing—original draft, Writing—review and editing; Xavier Belles, Conceptualization, Resources, Supervision, Methodology, Writing—review and editing; Coby Schal, Conceptualization, Resources, Investigation, Methodology, Writing—review and editing; Ayako Wada-Katsumata, Investigation, Methodology; Matthew J Amicucci, Conceptualization, Investigation, Methodology, Writing—review and editing; Artyom Kopp, Conceptualization, Resources, Supervision, Funding acquisition, Writing—original draft, Project administration, Writing—review and editing

### Author ORCIDs

Judith Wexler (iD) https://orcid.org/0000-0003-1400-5411
Emily Kay Delaney (iD) https://orcid.org/0000-0003-3609-5702
Xavier Belles (iD) http://orcid.org/0000-0002-1566-303X
Coby Schal (iD) https://orcid.org/0000-0001-7195-6358
Matthew J Amicucci (iD) https://orcid.org/0000-0002-1392-9252
Artyom Kopp (iD) https://orcid.org/0000-0001-5224-0741

### Decision letter and Author response

Decision letter https://doi.org/10.7554/eLife.47490.040
Author response https://doi.org/10.7554/eLife.47490.041

# Additional files

## Supplementary files

• Supplementary file 1. Sources of arthropod gene models used in phylogenetic analyses of the DMRT and SR gene families. Orders are color-coded as follows: holometabolous insects (orange), hemimetabolous insects (green), non-insect hexapods (red), crustaceans (blue), and chelicerates (purple).

DOI: https://doi.org/10.7554/eLife.47490.023

• Supplementary file 2. Distances between *prospero* and *doublesex* in the genomes of 10 different insect species.

DOI: https://doi.org/10.7554/eLife.47490.024

• Supplementary file 3. Scaffolds containing *prospero* and their sizes from nine hemipteran species.

DOI: https://doi.org/10.7554/eLife.47490.025

• Supplementary file 4. Queries used in BLASTP searches of arthropod gene models for DMRT homologs.

DOI: https://doi.org/10.7554/eLife.47490.026

• Supplementary file 5. Queries used in BLASTP searches of arthropod gene models for Transformer and SR family proteins.

DOI: https://doi.org/10.7554/eLife.47490.027

• Supplementary file 6. Estimated percentages of cuticular hydrocarbons in wild-type and *dsBgTra*-treated *Blattella germanica*. Each gas chromatograph (GC) peak is represented as a percentage of the total amount of all 30 hydrocarbons. The peak numbers correspond to the CHCs identified by *Jurenka et al. (1989)*. Peak 15 (9-, 11-, 13-, and 15-methylnonacosane) is known as a male-enriched CHC, and peak 22 (3,7-, 3,9-, and 3,11-dimethylnonacosane) is a female-enriched CHC. 3,11-Dimethylnonacosane also serves as precursor to several components of the female contact sex pheromone.

DOI: https://doi.org/10.7554/eLife.47490.028

• Supplementary file 7. Primers used in RACE, RT-PCR, and qPCR in this study.

DOI: https://doi.org/10.7554/eLife.47490.029

• Transparent reporting form

DOI: https://doi.org/10.7554/eLife.47490.030

## Data availability

PacBio and Illumina data sets have been deposited in NCBI's SRA under the project PRJNA552102. mRNA sequences from all transcripts isolated have been submitted to NCBI's GenBank under the accession numbers MK919524–MK919542, inclusive, and MN339474 and MN339475.

The following dataset was generated:

| Author(s) | Year | Dataset title | Dataset URL | Database and Identifier |
|---|---|---|---|---|
| Judith Wexler, Emily Kay Delaney, Artyom Kopp | 2019 | Investigation into sex differentiation pathway of hemimetabolous insects | https://www.ncbi.nlm.nih.gov/bioproject/?term=PRJNA552102 | NCBI BioProject, PRJNA552102 |

The following previously published datasets were used:

| Author(s) | Year | Dataset title | Dataset URL | Database and Identifier |
|---|---|---|---|---|
| Poelchau M, Childers C, Moore G, Tsavatapalli V, Evans J, Lee Lin H, Hackett K | 2014 | i5K project | https://i5k.nal.usda.gov/content/data-downloads | USDA, Arthropoda |
| Giraldo-Calderón GI, Emrich SJ, MacCallum RM, Maslen G, Dialynas | 2015 | VectorBase: an updated bioinformatics resource for invertebrate vectors and other organisms related with human | https://www.vectorbase.org/downloads?field_organism_taxonomy_tid%5B%5D=398&field_ | Pediculus humanus, PhumU2 |

| | | | | |
|---|---|---|---|---|
| E, Topalis P, Ho N, Gesing S, Vector-Base Consortium, Madey G, Collins FH, Lawson D | | diseases | download_file_type_tid%5B%5D=412&field_download_file_format_tid=All&field_status_va-lue=Current | |
| Giraldo-Calderón GI, Emrich SJ, MacCallum RM, Maslen G, Dialynas E, Topalis P, Ho N, Gesing S, Vector-Base Consortium, Madey G, Collins FH, Lawson D | 2015 | VectorBase: an updated bioinformatics resource for invertebrate vectors and other organisms related with human diseases | https://www.vectorbase.org/downloads?field_organism_taxonomy_tid%5B%5D=393&field_download_file_type_tid%5B%5D=412&field_download_file_format_tid=All&field_status_va-lue=Current | Rhodnius prolixus, RproC3 |

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
