## [Decision Letter]

Thank you for submitting your article "Hemimetabolous insects elucidate the origin of sexual development via alternative splicing" for consideration by *eLife*. Your article has been reviewed by three peer reviewers, and the evaluation has been overseen by a Reviewing Editor and Patricia Wittkopp as the Senior Editor. The following individual involved in review of your submission has agreed to reveal their identity: Michael Perry (Reviewer #3).

The reviewers have discussed the reviews with one another and the Reviewing Editor has drafted this decision to help you prepare a revised submission.

The three reviewers closely agree that this is an important paper that should be published. However, the reviewers also find that the paper could be seriously improved if the logic and the writing were to be improved in several sections as the overall message is diluted by several difficult sections.

Although one reviewer suggested that adding more species might help, it is likely that more particular cases would be found and this would not clarify the message as they might not represent an overarching trajectory of evolution toward *Drosophila*-like sex determination. However, we would like to encourage you to address this issue in your Discussion.

Reviewer #1:

The premise for this study is to understand how sex determining mechanisms are both rapidly evolving and have conservation of the downstream genes specifying sexual development, with a focus on insect sex determining pathways. In holometabolous insects there is a critical regulatory cassette that includes two genes transformer and *dsx. tra* and *dsx* encode transcripts that are sex-specifically spliced to produce sex-specific proteins, with *tra* regulating *dsx* splicing. Ancestrally, *tra* and *dsx* are regulated independently of each other and male-required *dsx* is regulated transcriptionally for sexual development.

This study examines *tra* and *dsx* in three hemimetabolous insects to bridge a gap in current knowledge of sex determination mechanisms, to understand the potential evolutionary transitions that give rise to different mechanisms. While it is now clear that sex determination evolves rapidly, the molecular-genetic mechanisms that change are not understood. This study provides compelling evidence regarding novel and conserved mechanisms for *tra* and *dsx* regulation in three species examined. The mechanistic roles of *tra* and *dsx* in sexual development in the German cockroach (*Blattella germanica*) are more deeply examined using RNAi approaches. Overall, the study is carefully performed, using several molecular and computational approaches to identify and validate *tra* and *dsx* transcript isoforms from three species. The results of the functional studies in *Blattella* are unexpected and lead to interesting insights about evolution of gene/transcript regulation and function. The study also provides insights into the transitions that might lead to novel sex determination genetic pathways during evolution.

1) The identification of *tra* and *dsx* transcripts is a critical aspect of the study. It appears that the authors were thorough in their search for all transcript isoforms encoded by *tra* and *dsx*, in the three species examined. The gene models presented are convincing and carefully annotated. Can the authors provide more rationales/justification for the tissues they used to identify cDNAs, in terms of identification of all transcript isoforms. For example, can the authors find all transcript isoforms for *tra* and *dsx* in public RNA-seq data sets from *Drosophila melanogaster* by searching data sets made from the same tissues. Are there other data sets that justify the use of the tissues chosen here? Is there an alternative justification that they can provide? The concern is that the tissues used were gonadal (somatic and germline tissues), as well as additional somatic tissues, so will contain a high abundance of transcripts from germline-expressed genes. *tra* and *dsx* are genes with functions in somatic sex determination. Also, please clarify if independent RNA samples were used to validate the cDNAs identified in the three different species.

2) The other critical findings are from the RNAi knockdown experiments in *Blatella*.

A) The authors should provide a rationale as to why they chose to perform RNAi experiments in this species and not the other two. This was not clear in the paper.

B) The unexpected results are that RNAi knockdown of *tra*, but not *dsx*, results in female sex transformation/developmental phenotypes. This is surprising because the authors show that *tra* does regulate *dsx* splicing and a male form of *dsx* is even apparent in *tra* RNAi knockdown females. On the other hand, RNAi knockdown of *dsx* causes male developmental phenotypes, but not RNAi knockdown of *tra*.

It appears that *tra* RNAi is effective in females, given appearance of male *dsx* transcripts and sex-transformation phenotypes in morphology and gene expression. The impact of *tra* RNAi in males is less clear, given there are no morphological phenotypes and the reduction in expression is more modest than it appears in females (~2.3 vs. ~18 fold reduction; Figure 3—figure supplement 1). The RNA blot analyses in Figure 6 show that dsTraRNAi in females results in presence of male-specific *dsx* transcript isoforms, but the authors did not look for female *dsx* isoforms in dsTraRNAi males. Additionally, in Figure 4 and other related analyses the wild type female control is not used.

Additionally, the authors tried, but were not able to show that *dsx* RNAi is effective in either males or females, using qPCR. However, the *dsx* RNAi is effective in males, but not females, so there is nothing wrong with the reagents or the targeted regions in the *dsx* transcript-isoforms.

Overall, I think the *Blatella* mechanistic studies were performed well, given how challenging the experiments are. For both sets of experiments the authors have positive results using their RNAi reagents. However, given some of the criticisms noted above, the authors may want to comment on these points to further clarify and strengthen their conclusions.

Some of the methodology for the RNAi injections are not clear. For example, in some cases it is not clear where the RNAi molecules are injected into the animal in each experiment. How many times were injections performed? Primer pairs for qPCR to detect *tra* after RNAi knockdown are not shown in Figure 1 or the supplementary figure.

3) I think the presentation would be easier to follow if the phylogenetic trees were presented earlier than Figure 8 – perhaps a supplementary figure would work and Figure 8 can remain. As I read the paper, I was not aware of Figure 8, so searched for these trees. Also, a more detailed summary figure/table with the results from the three species examined here would help keep track of the results. It might help a reader outside of the sex determination field to provide a figure with the holometabolous insect sex determination pathway. Perhaps the authors could also add a model in the supplement for the proposed roles of *tra* and *dsx* in Blatella, given the unexpected findings.

Reviewer #2:

This paper is important because it aims to tackle the origin of the insect-specific sex-determination mechanism that involves sex-specific splicing of the doublesex mRNA to produce sex-specific transcription factors. Existing information about sex-determining systems has a big gap between crustaceans and holometabolous insects. This gap exists because previous analyses have been largely limited to model organisms and to species some researchers happened to be interested in. This paper is notable in choosing species that are best positioned to answer the evolutionary question at hand, which is the origin of the *Drosophila*-like system. Answering this question not only is a contribution to the understanding of sex determination, but also contributes to greater understanding of the evolution of developmental processes in general.

The authors investigate three hemimetabolous insect species: the kissing bug, the louse and the German cockroach. They find that doublesex is sex-specifically spliced in all three, but that only the kissing bug shows *Drosophila*-like sex-specific splicing of transformer. In addition, they find that the cockroach doublesex, although sex-specifically spliced, is only required for male sexual differentiation. These results suggest that the evolution of this splicing cascade has been gradual, with sex-specific splicing of doublesex arising while doublesex maintained the ancestral male-specific role.

The sequence analyses and experiments appear to have been conducted rigorously and with appropriate care. The usual worries in such analyses are: 1) have orthologs been properly identified? and 2) are RNAi phenotypes specific (and are there no false negatives)? The authors have considered these nuances and, to me, have done a convincing job in interpreting their results and are candid about the limitations of this kind of comparative work (especially with respect to the fast-evolving, disordered Tra protein). But especially the results with the cockroach are hard to explain except as knockdowns of true orthologs.

My only hesitation about this work is that it only involves three species. Given the pecularities/secondary losses/etc. seen in this pathway in the species that have been studied so far, it is hard to take any one species as representative of an entire lineage. That said, finding male-specific function of *dsx* and female-specific function of *tra* in cockroach is revealing and important, and the other two species show additional steps along the way to Holometabola (albeit with some uncertainty as to when given the uncertainty of the insect phylogeny), as well summarized in Figure 8. One suggestion I would have would be to expand Figure 8 to include other transitions that are known from the literature (e.g., within Holometabola including CAM, Sxl, etc.). The reason I suggest this is that this paper is likely to become a touchstone for the field, and summarizing all of the information in one place would be very useful to others.

Reviewer #3:

In this manuscript the authors investigate the role of Transformer and Doublesex in sex determination in three basal insect clades using transcriptomic analysis as well as functional tools in the Blattodea (cockroach). This work uncovers several interesting differences that suggest possible intermediate states between the higher insects (largely represented by work in *Drosophila*) and a much less well-studied ancestral state represented by data from a handful of papers in various outgroups. Though the results suggest a somewhat complicated evolutionary history with possible secondary reversions and group-specific specializations, the authors are able to suggest a coherent model for the shift from transcription-based differences in *Dsx* expression to splicing-based differences via Tra. Overall, the work represents a significant advance over what was known previously by examining several difficult to work with groups, and these efforts will lay the foundation for much future work on the evolution of this fascinating system.

I have only a few relatively minor concerns and specific suggestions (below). Perhaps the biggest question I am left with is how *Dsx* is differentially spliced in B. germanica when both sexes produce the same isoforms of Tra. The Discussion does a good job laying out possibilities but these are difficult to test and it would be great to know what else is targeted by Tra or that interacts with Tra to control sex. Unfortunately, I can't think of an easy or simple experiment to sort this out!

Given the speculation about the variety of isoforms it is tempting to start thinking about experiments that might test the ideas proposed. What would happen if the *BgTra* isoform with the "interrupting exon" in the CAM domain were specifically knocked down? The section that describes how the interrupting exon relates to the evolution of a male-specific stop codon is not especially clear (subsection “The role of *transformer* in sexual differentiation predates canonical sex-specific splicing of *tra*”, last paragraph). This section should at least be revised, even if further isoform-specific knock down is beyond the scope of the current paper.

While I am not especially surprised that Tra plays the role that it does in Blattodea (complete control of sexually specific morphology and probably behavior as well) this was previously not demonstrated and has been done well here. The differences in *Dsx* splicing and expression are unfortunately complicated but also well documented. I especially appreciate the review of current literature on specification of sex in not well studied insect outgroups, and the placement of the new findings in this context.

---

## [Author Response]

The three reviewers closely agree that this is an important paper that should be published. However, the reviewers also find that the paper could be seriously improved if the logic and the writing were to be improved in several sections as the overall message is diluted by several difficult sections.Although one reviewer suggested that adding more species might help, it is likely that more particular cases would be found and this would not clarify the message as they might not represent an overarching trajectory of evolution toward Drosophila-like sex determination. However, we would like to encourage you to address this issue in your Discussion.

We agree – both that additional taxon sampling will be beneficial in the long term, and that adding new taxa is likely to uncover new lineage-specific changes. Our understanding of sexual differentiation in holometabolous insects rests on studies of about 15 Dipteran species, and about 10 non-Dipteran insects, described in >50 papers. Given the age and diversity of hemimetabolous orders, we expect it will require at least as many taxa to fully understand the sequence of evolutionary changes leading up to the origin of Holometabola. We hope that our report will help inform the design of future studies in this area.

In the revised manuscript, we added the following sentences in the Discussion: “A more extensive sampling of hemimetabolous insects will no doubt be necessary to reconstruct the full series of events that produced the familiar holometabolous mode of sexual differentiation, and to determine the timing of the key synapomorphies among the variety of lineage-specific changes. Based on our limited taxon sampling, we can propose a tentative model.” [Followed by the description of our model reflected in Figure 8]. We then end our paper with the following sentence: “Comparative and functional work in other basal insects, non-insect Hexapods, and crustaceans will be needed to determine the ancestral functions of *tra*, the roles of female-specific *dsx* isoforms in hemimetabolous insects, and other details of this emerging model.”.

Reviewer #1:The premise for this study is to understand how sex determining mechanisms are both rapidly evolving and have conservation of the downstream genes specifying sexual development, with a focus on insect sex determining pathways. In holometabolous insects there is a critical regulatory cassette that includes two genes transformer and dsx. tra and dsx encode transcripts that are sex-specifically spliced to produce sex-specific proteins, with tra regulating dsx splicing. Ancestrally, tra and dsx are regulated independently of each other and male-required dsx is regulated transcriptionally for sexual development.This study examines tra and dsx in three hemimetabolous insects to bridge a gap in current knowledge of sex determination mechanisms, to understand the potential evolutionary transitions that give rise to different mechanisms. While it is now clear that sex determination evolves rapidly, the molecular-genetic mechanisms that change are not understood. This study provides compelling evidence regarding novel and conserved mechanisms for tra and dsx regulation in three species examined. The mechanistic roles of tra and dsx in sexual development in the German cockroach (Blattella germanica) are more deeply examined using RNAi approaches. Overall, the study is carefully performed, using several molecular and computational approaches to identify and validate tra and dsx transcript isoforms from three species. The results of the functional studies in Blattella are unexpected and lead to interesting insights about evolution of gene/transcript regulation and function. The study also provides insights into the transitions that might lead to novel sex determination genetic pathways during evolution.1) The identification of tra and dsx transcripts is a critical aspect of the study. It appears that the authors were thorough in their search for all transcript isoforms encoded by tra and dsx, in the three species examined. The gene models presented are convincing and carefully annotated. Can the authors provide more rationales/justification for the tissues they used to identify cDNAs, in terms of identification of all transcript isoforms. For example, can the authors find all transcript isoforms for tra and dsx in public RNA-seq data sets from Drosophila melanogaster by searching data sets made from the same tissues. Are there other data sets that justify the use of the tissues chosen here? Is there an alternative justification that they can provide? The concern is that the tissues used were gonadal (somatic and germline tissues), as well as additional somatic tissues, so will contain a high abundance of transcripts from germline-expressed genes. tra and dsx are genes with functions in somatic sex determination.

First, a quick recap: For the cockroach, we used fat body, gonads, and accessory glands. For *Rhodnius*, we used fat body, gonads, and internal reproductive tracts. For louse, we used whole adults.

We used these tissues for two reasons: an a priori expectation that, since the gonads and reproductive tracts are sex-specific, they are likely to express the genes required for sexual differentiation; and our knowledge of gene expression in *Drosophila*. In flies, *dsx* is transcribed, and spliced sex-specifically, in the somatic gonads and in the fat body (among other tissues), while *tra* expression is apparently ubiquitous (Camara et al., 2019; Clough et al., 2014). Fat body is also known to have a high prevalence of sex-biased gene expression that is regulated by *dsx* and *tra* (Arbeitman et al., 2016; Chang et al., 2011). Little is known about the tissue-specificity of *dsx* and *tra* expression in non-Dipteran insects, but *Dsx* protein is expressed in the pupal wings of butterflies that have sex-specific wing color patterns (Kunte et al., 2014), supporting the idea that tissues that go through sexually dimorphic differentiation are likely to express *dsx*. It is true that gonadal tissues contain a high proportion of germline cells, and that the *Drosophila dsx* and *tra* are involved in somatic but not in germline sexual differentiation. However, we do not see this as a concern because (1) our cloning and transcriptome sequencing strategies do not require *all* cells in our tissue samples to be expressing *dsx* and *tra*, and (2) we actually do not know that *dsx* and *tra* are NOT involved in germline sexual differentiation in non-Dipteran insects, so we saw the inclusion of sex-specific germline tissues in our samples as an insurance strategy. (As a side note, the presence of separate mechanisms for somatic and germline sex determination that we have in *Drosophila* is unusual even among Diptera (Murray et al., 2010)).

To address the reviewer’s specific question: the male-specific and female-specific isoforms of *dsx* and *tra* are present in *Drosophila* RNA-seq datasets representing the gonads, fat body, and reproductive organs.

There are two reported isoforms of *tra* in *Drosophila melanogaster* – the male- and the female-specific one. According to gene expression data collated on Fly Atlas, these isoforms are both present in the fat body of *D. melanogaster* males and females, respectively (http://flyatlas.gla.ac.uk/FlyAtlas2/index.html?search=gene&gene=tra&idtype=symbol#mobileTargetG).

There are three male and three female isoforms of *D. melanogaster dsx*. The female isoforms are all identical in their coding sequence; two of the male isoforms are identical in their coding sequence, while the third has an additional 23 amino acids at the C-terminus. According to the data collated on Fly Atlas, one of the male isoforms is present in the accessory glands, and one in the fat body. The *dsxM* isoform not represented in the Fly Atlas dataset has the same coding sequence as the *dsxM* isoform expressed in the fat body. Two of the three *dsxF* isoforms are present in the fat body. The third isoform is not expressed in any tissues homologous to those we studied, but its coding sequence does not differ from the two other *dsxF* isoforms with reported expression. Based on many years of *Drosophila* community resource development, including deep sequencing and microarray analysis of multiple tissues and developmental stages and extensive sequencing of cloned EST libraries, only 3 of the *dsx* isoforms are reasonably common; the rest were only identified recently through ultra-deep sequencing, and presumably represent rare isoforms.

Extrapolating from the *Drosophila* experience, we believe that although our sequencing efforts may have missed some rare isoforms, we have most likely captured the major *dsx* and *tra* isoforms by sequencing the main sexually dimorphic tissues. In the revised manuscript, we added the following sentence to Materials and methods: “We reasoned that sexually dimorphic tissues were likely to express *doublesex* and *transformer*, including any sex-specific isoforms of these genes that might exist.”. In the Results section, the relevant sentence has been changed to “To understand when *tra*-dependent alternative splicing of *dsx* evolved, we identified *tra* isoforms expressed in sexually dimorphic tissues of males and females from three hemimetabolous insect orders…” and “To test for the presence of sex-specific *dsx* isoforms, we isolated *dsx* transcripts using PacBio Isosequencing, Illumina RNA-sequencing, and 5’ and 3’ RACE on sexually dimorphic tissues.”

Also, please clarify if independent RNA samples were used to validate the cDNAs identified in the three different species.

Yes, they were. For *Pediculus* and *Rhodnius*, we stated that different RNA samples were used for RACE (to identify *tra* and *dsx* isoforms) and for rt-PCR (to test for sex-specificity of these isoforms) (subsection “Reverse-transcription PCR tests of isoform sex-specificity”). In the revised manuscript, we have also added the following sentence for *Blattella*:

“These RNA samples were isolated from individuals that were not used to construct PacBio or RACE libraries.”

2) The other critical findings are from the RNAi knockdown experiments in Blatella.A) The authors should provide a rationale as to why they chose to perform RNAi experiments in this species and not the other two. This was not clear in the paper.

For RNAi, we had to be able to culture the insects in our lab through their complete life cycle. This was possible for *Blattella*, but not for *Rhodnius* and *Pediculus*. The latter two species are obligate blood-feeders as well as human pathogen vectors; special biocontainment facilities and permits are required for their maintenance. Despite these disadvantages, we chose *Pediculus humanus* because Phthiraptera occupy a key phylogenetic position vis-à-vis the Holometabolous insects, and *P. humanus* was the easiest representative of Phthiraptera to obtain (all Phthiraptera are parasitic – so we might as well use the one with the most abundant host). We chose *R. prolixus* because it had by far the most convincing *dsx* ortholog among Hemiptera, and orthology was clearly essential to our study. *R. prolixus* samples were sent to us from Canada by Ian Orchard (at the time, no lab in the US had a *Rhodnius* permit). Louse samples were collected from the local human population and processed immediately for RNA extraction.

In the revised manuscript, we added the following sentences to Materials and methods:

“RNAi experiments were conducted in *B. germanica* due to the ease of culturing this insect. In contrast, *R. prolixus* and *P. humanus* are obligate blood feeders, making them difficult to culture outside of specialized biocontainment facilities.”.

B) The unexpected results are that RNAi knockdown of tra, but not dsx, results in female sex transformation/developmental phenotypes. This is surprising because the authors show that tra does regulate dsx splicing and a male form of dsx is even apparent in tra RNAi knockdown females. On the other hand, RNAi knockdown of dsx causes male developmental phenotypes, but not RNAi knockdown of tra.It appears that tra RNAi is effective in females, given appearance of male dsx transcripts and sex-transformation phenotypes in morphology and gene expression.

Yes, this is an accurate summary of the results.

The impact of tra RNAi in males is less clear, given there are no morphological phenotypes and the reduction in expression is more modest than it appears in females (~2.3 vs. ~18 fold reduction; Figure 3—figure supplement 1).

To put this in perspective, there is much less *tra* in wild-type males compared to wild-type females, so there is less scope for knock-down. In normalized units, *tra* expression is reduced from ~450 to ~20 in females, and from ~35 to ~15 in males (Figure 3—figure supplement 1). The fact that *tra* is not required for male sexual differentiation is one of the similarities between roach and *Drosophila*.

The RNA blot analyses in Figure 6 show that dsTraRNAi in females results in presence of male-specific dsx transcript isoforms, but the authors did not look for female dsx isoforms in dsTraRNAi males.

We have performed rtPCR for *dsx* on *dsTraRNAi* males. We found that these *tra*-depleted males were similar to wild-type males, in that they still expressed the male-specific *dsx* isoform and did not express a detectable amount of the female-specific *dsx* isoform. This result is in agreement with what we know from the holometabolous insects, where *tra* is necessary for the production of the female-specific, but not male-specific *dsx* isoforms. In the revised manuscript, we added the following sentences to the Results: “As expected, no change in *BgDsx* splicing was observed in *dsBgTra* males. This is similar to the pattern of *tra-*dependent female *dsx* splicing seen in holometabolous insects…”.

Additionally, in Figure 4 and other related analyses the wild type female control is not used.

We had collected wild-type female data as part of the original experiments. In the revised manuscript, we added these data to Figure 4 to better illustrate the point that *dsBgTra* females act more like wild-type males than like wild-type females (which neither respond to stimulation with female antennae nor elicit a feeding response in females). We also revised Figure 4 legend accordingly. For the tergal gland analysis (Figure 4—figure supplement 1), we could not test wild-type females since they have no structure analogous to a tergal gland on which we could perform a chemical assays.

Additionally, the authors tried, but were not able to show that dsx RNAi is effective in either males or females, using qPCR. However, the dsx RNAi is effective in males, but not females, so there is nothing wrong with the reagents or the targeted regions in the dsx transcript-isoforms.Overall, I think the Blatella mechanistic studies were performed well, given how challenging the experiments are. For both sets of experiments the authors have positive results using their RNAi reagents. However, given some of the criticisms noted above, the authors may want to comment on these points to further clarify and strengthen their conclusions.Some of the methodology for the RNAi injections are not clear. For example, in some cases it is not clear where the RNAi molecules are injected into the animal in each experiment. How many times were injections performed?

As described in Materials and methods, we injected both nymphs and adults in the abdomen between two sternites. For *doublesex*, we injected nymphs with three different constructs in three different experiments (n = 8 – 15 females or males per experiment). Two of the constructs targeted the DM domain; one targeted the *dsx* dimerization domain (Figure 2C). To look at the effect of *dsx* on embryogenesis, we injected adult females in two separate trials (n=11 and =8), using the same dsRNA construct (targeting the DM domain) for both. For *transformer*, we injected male and female nymphs with two different RNAi constructs in two separate experiments (n =15 per trial). We performed three separate experiments with adult females to observe the effect of *tra* on embryogenesis, using two different constructs. Trials for the maternal transformer RNAi had 8-17 females per experiment.

In the revised manuscript, we added these details to Materials and methods (subsection “RNAi injections, dissections and imaging”, first paragraph).

Primer pairs for qPCR to detect tra after RNAi knockdown are not shown in Figure 1 or the supplementary figure.

We have added a new panel to Figure 3—figure supplement 1 to show the location of the primers used to check *tra* abundance, and expanded legend accordingly.

3) I think the presentation would be easier to follow if the phylogenetic trees were presented earlier than Figure 8 – perhaps a supplementary figure would work and Figure 8 can remain. As I read the paper, I was not aware of Figure 8, so searched for these trees.

This is an excellent point. We have expanded Figure 1—figure supplement 1 to include a simple schematic of the arthropod phylogeny including the hemimetabolous orders we studied. Note that we also combined the old Supplementary Figures 1 and 2 into the new Figure 1—figure supplement 1.

Also, a more detailed summary figure/table with the results from the three species examined here would help keep track of the results. It might help a reader outside of the sex determination field to provide a figure with the holometabolous insect sex determination pathway. Perhaps the authors could also add a model in the supplement for the proposed roles of tra and dsx in Blatella, given the unexpected findings.

Also a very good suggestion. In the revised manuscript, we added a new panel to Figure 8 to summarize our results in comparison to the two “bookends” (Holometabola and *Daphnia*), and expanded Figure 8 legend accordingly.

Reviewer #2:This paper is important because it aims to tackle the origin of the insect-specific sex-determination mechanism that involves sex-specific splicing of the doublesex mRNA to produce sex-specific transcription factors. Existing information about sex-determining systems has a big gap between crustaceans and holometabolous insects. This gap exists because previous analyses have been largely limited to model organisms and to species some researchers happened to be interested in. This paper is notable in choosing species that are best positioned to answer the evolutionary question at hand, which is the origin of the Drosophila-like system. Answering this question not only is a contribution to the understanding of sex determination, but also contributes to greater understanding of the evolution of developmental processes in general.The authors investigate three hemimetabolous insect species: the kissing bug, the louse and the German cockroach. They find that doublesex is sex-specifically spliced in all three, but that only the kissing bug shows Drosophila-like sex-specific splicing of transformer. In addition, they find that the cockroach doublesex, although sex-specifically spliced, is only required for male sexual differentiation. These results suggest that the evolution of this splicing cascade has been gradual, with sex-specific splicing of doublesex arising while doublesex maintained the ancestral male-specific role.The sequence analyses and experiments appear to have been conducted rigorously and with appropriate care. The usual worries in such analyses are: 1) have orthologs been properly identified? and 2) are RNAi phenotypes specific (and are there no false negatives)? The authors have considered these nuances and, to me, have done a convincing job in interpreting their results and are candid about the limitations of this kind of comparative work (especially with respect to the fast-evolving, disordered Tra protein). But especially the results with the cockroach are hard to explain except as knockdowns of true orthologs.My only hesitation about this work is that it only involves three species. Given the pecularities/secondary losses/etc. seen in this pathway in the species that have been studied so far, it is hard to take any one species as representative of an entire lineage. That said, finding male-specific function of dsx and female-specific function of tra in cockroach is revealing and important, and the other two species show additional steps along the way to Holometabola (albeit with some uncertainty as to when given the uncertainty of the insect phylogeny), as well summarized in Figure 8.

Given the deep divergence of hemimetabolous insect orders, we expect that additional taxon sampling will uncover not only general trends in the evolution of sexual development, but also many lineage-specific peculiarities. For example, our inability to find sex-specific isoforms of *dsx* in the louse almost certainly reflects a secondary loss of sexually dimorphic splicing. On the other hand, the fact that we see a previously undescribed mode of *tra* splicing -- one in which the CAM domain is interrupted by an exon -- in two lineages separated by over 350 million years (*Blattella* and *Pediculus*) is evidence that even with limited taxon sampling some general mechanisms can be identified. No doubt more will be learned in the future by conducting functional analyses in additional species. We hope this report will help guide new studies in this area.

In the revised manuscript, we added the following sentences in the Discussion: “A more extensive sampling of hemimetabolous insects will no doubt be necessary to reconstruct the full series of events that produced the familiar holometabolous mode of sexual differentiation, and to determine the timing of the key synapomorphies among the variety of lineage-specific changes. Based on our limited taxon sampling, we can propose a tentative model.” [Followed by the description of our model reflected in Figure 8]. We then end our paper with the following sentence: “Comparative and functional work in other basal insects, non-insect Hexapods, and crustaceans will be needed to determine the ancestral functions of *tra*, the roles of female-specific *dsx* isoforms in hemimetabolous insects, and other details of this emerging model.”

One suggestion I would have would be to expand Figure 8 to include other transitions that are known from the literature (e.g., within Holometabola including CAM, Sxl, etc.). The reason I suggest this is that this paper is likely to become a touchstone for the field, and summarizing all of the information in one place would be very useful to others.

The differences we observe among hemimetabolous insects are in contrast to the relative stasis seen in Holometabola. To a first approximation, the *tra-dsx* relationship is similar in Hymenoptera, Coleoptera, and Diptera. In Holometabola, the secondary loss of the CAM domain of Tra is specific to Drosophilidae, as is the role of *Sxl* in sex determination (even in other Cyclorrhaphan Dipterans, *Sxl* does not appear to be involved in sexual development). Given the recency of these events compared to the phylogenetic scale of changes summarized in Figure 8A-B, we think it is better not to include these *Drosophila* idiosyncrasies into the phylogenetic scheme. In what we hope is an alternative solution, we added a new panel to Figure 8 (Figure 8C) that summarizes our results from hemimetabolous insect orders in comparison to holometabolous insects and crustaceans. In the revised manuscript, we also expanded Figure 8 legend to describe the evolutionary changes relating to the CAM domain and *Sxl*, as suggested by the reviewer.

Reviewer #3:In this manuscript the authors investigate the role of Transformer and Doublesex in sex determination in three basal insect clades using transcriptomic analysis as well as functional tools in the Blattodea (cockroach). This work uncovers several interesting differences that suggest possible intermediate states between the higher insects (largely represented by work in Drosophila) and a much less well-studied ancestral state represented by data from a handful of papers in various outgroups. Though the results suggest a somewhat complicated evolutionary history with possible secondary reversions and group-specific specializations, the authors are able to suggest a coherent model for the shift from transcription-based differences in Dsx expression to splicing-based differences via Tra. Overall, the work represents a significant advance over what was known previously by examining several difficult to work with groups, and these efforts will lay the foundation for much future work on the evolution of this fascinating system.I have only a few relatively minor concerns and specific suggestions (below). Perhaps the biggest question I am left with is how Dsx is differentially spliced in B. germanica when both sexes produce the same isoforms of Tra. The Discussion does a good job laying out possibilities but these are difficult to test and it would be great to know what else is targeted by Tra or that interacts with Tra to control sex. Unfortunately, I can't think of an easy or simple experiment to sort this out!Given the speculation about the variety of isoforms it is tempting to start thinking about experiments that might test the ideas proposed. What would happen if the BgTra isoform with the "interrupting exon" in the CAM domain were specifically knocked down?

These are both excellent questions. However, answering these questions definitively will require extensive new experiments in *B. germanica* that we are not yet equipped to perform, so we only have possible hypotheses at this point. We have described these hypotheses in the Discussion. With the data we have, we would not feel comfortable speculating beyond this.

The section that describes how the interrupting exon relates to the evolution of a male-specific stop codon is not especially clear (subsection “The role of transformer in sexual differentiation predates canonical sex-specific splicing of tra”, last paragraph). This section should at least be revised, even if further isoform-specific knock down is beyond the scope of the current paper.

We have re-written that section of the Discussion (subsection “The role of transformer in sexual differentiation predates canonical sex-specific splicing of *tra*”, last paragraph). We hope that the revised version conveys our hypothesis more clearly.

While I am not especially surprised that Tra plays the role that it does in Blattodea (complete control of sexually specific morphology and probably behavior as well) this was previously not demonstrated and has been done well here. The differences in Dsx splicing and expression are unfortunately complicated but also well documented. I especially appreciate the review of current literature on specification of sex in not well studied insect outgroups, and the placement of the new findings in this context.